# Controlled nitric oxide production via $O(^1D)$+$N_2O$ reactions for use in oxidation flow reactor studies

Andrew Lambe[1,2], Paola Massoli[1], Xuan Zhang[1,*], Manjula Canagaratna[1], John Nowak[1,**],
Conner Daube[1], Chao Yan[3], Wei Nie[4,3], Timothy Onasch[1,2], John Jayne[1], Charles Kolb[1],
Paul Davidovits[2], Douglas Worsnop[1,3], and William Brune[5]

[1]Aerodyne Research, Inc., Billerica, Massachusetts, United States
[2]Chemistry Department, Boston College, Chestnut Hill, Massachusetts, United States
[3]Physics Department, University of Helsinki, Helsinki, Finland
[4]Joint International Research Laboratory of Atmospheric and Earth System Sciences, School of Atmospheric Sciences, Nanjing University, Nanjing, China
[5]Department of Meteorology and Atmospheric Sciences, The Pennsylvania State University, University Park, Pennsylvania, United States
[*]Current address: Atmospheric Chemistry Observations & Modeling Laboratory, National Center for Atmospheric Research, Boulder, Colorado, United States
[**]Current address: Chemistry and Dynamics Branch, NASA Langley Research Center, Hampton, Virginia, United States

*Correspondence to:* Andrew Lambe (lambe@aerodyne.com), William Brune (whb2@psu.edu)

**Abstract.**

Oxidation flow reactors that use low-pressure mercury lamps to produce hydroxyl (OH) radicals are an emerging technique for studying the oxidative aging of organic aerosols. Here, ozone ($O_3$) is photolyzed at 254 nm to produce $O(^1D)$ radicals, which react with water vapor to produce OH. However, the need to use parts-per-million levels of $O_3$ hinders the ability of oxidation flow reactors to simulate $NO_x$-dependent SOA formation pathways. Simple addition of nitric oxide (NO) results in fast conversion of $NO_x$ (NO + $NO_2$) to nitric acid ($HNO_3$), making it impossible to sustain $NO_x$ at levels that are sufficient to compete with hydroperoxy ($HO_2$) radicals as a sink for organic peroxy ($RO_2$) radicals. We developed a new method that is well suited to the characterization of $NO_x$-dependent SOA formation pathways in oxidation flow reactors. NO and $NO_2$ are produced via the reaction $O(^1D) + N_2O \rightarrow 2NO$, followed by the reaction $NO + O_3 \rightarrow NO_2 + O_2$. Laboratory measurements coupled with photochemical model simulations suggest that $O(^1D) + N_2O$ reactions can be used to systematically vary the relative branching ratio of $RO_2$ + NO reactions relative to $RO_2$ + $HO_2$ and/or $RO_2$ + $RO_2$ reactions over a range of conditions relevant to atmospheric SOA formation. We demonstrate proof of concept using high-resolution time-of-flight chemical ionization mass spectrometer (HR-ToF-CIMS) measurements with nitrate ($NO_3^-$) reagent ion to detect gas-phase oxidation products of isoprene and $\alpha$-pinene previously observed in $NO_x$-influenced environments and in laboratory chamber experiments.

## 1 Introduction

Recent atmospheric observations supported by experimental and theoretical studies show that highly oxidized molecules (HOM), together with sulfuric acid, are involved in the initial nucleation steps leading to new particle formation (NPF)

(Donahue et al., 2013; Riccobono et al., 2014; Kurten et al., 2016). HOM form rapidly in the gas phase via auto-oxidation processes (Crounse et al., 2013; Rissanen et al., 2014) and tend to condense irreversibly (Ehn et al., 2014). Following NPF, semivolatile organic compounds (SVOC) with higher vapor pressures condense on newly formed aerosols at rates influenced by their volatility (Donahue et al., 2012), ultimately driving nanoparticle growth towards formation of cloud condensation nuclei (CCN) (Pierce et al., 2012; Riipinen et al., 2012). NPF events may produce as much as 50% of global CCN (Merikanto et al., 2009; Yu and Luo, 2009). However, mechanisms that govern the formation of specific HOM and condensation of SVOCs in various source regions are largely unknown.

The extent to which NPF and growth is influenced by natural and anthropogenic emissions, separately and together, is still unknown. In some locations, biogenic SOA formation is enhanced by anthropogenic carbonaceous aerosol particles, $SO_x$ and/or $NO_x$ (Carlton et al., 2010; Shilling et al., 2013; Xu et al., 2015). At the moment, one can only speculate about some of the possible synergistic or antagonistic chemical mechanisms regulating these processes. For example, anthropogenic emissions can enhance biogenic SOA formation by providing seed particles for condensable biogenic vapors. On the other hand, isoprene can slow down the formation of SOA from other volatile organics, possibly by depleting the local concentration of OH without itself producing significant SOA yields (Pugh et al., 2011). Globally the source strength of anthropogenic SOA is poorly constrained, with an uncertainty of at least a factor of 2 or 3 (Spracklen et al., 2011). Large uncertainties in pre-industrial aerosol emissions and processes further confound our understanding of the direct and indirect effects of anthropogenic aerosol emissions (Carslaw et al., 2014) and the impact of aerosols on climate (Andreae and Gelencsér, 2006).

To date, environmental chamber experiments have generated most of the laboratory SOA yield data used in atmospheric models, especially in simulations of polluted atmospheric conditions with elevated $NO_x$ concentrations. However, $NO_x$-dependent chambers studies are complicated by the need to use multiple OH radical precursors such as hydrogen peroxide ($H_2O_2$) and nitrous acid (HONO) or methyl nitrite ($CH_3ONO$) to span the relevant range of $NO_x$ levels (typically, $H_2O_2$ for low-$NO_x$ conditions and HONO or $CH_3ONO$ for high-$NO_x$ conditions) (Ng et al., 2007). Additionally, chambers have relatively low throughput and are limited to residence times of several hours due to chamber deflation and/or loss of particles and oxidized vapors to the chamber walls (Zhang et al., 2014). This restricts environmental chambers to simulating atmospheric aerosol particle lifetimes and SOA yields only up to 1 or 2 days, therefore limiting the study of formation of highly oxygenated SOA characteristic of aged atmospheric organic aerosol PM (Ng et al., 2010) unless very low VOC precursor concentrations are used (Shilling et al., 2009; Pfaffenberger et al., 2013).

Oxidation flow reactors have recently been developed to study SOA formation and evolution over time scales ranging from hours to multiple days of equivalent atmospheric OH exposure. In these reactors, $O_3$ is photolyzed at 254 nm to produce $O(^1D)$ radicals, which react with water vapor to produce OH radicals. OH concentrations are typically $10^8$ $cm^{-3}$ or greater. Under these conditions, atmospheric photochemical aging timescales up to ~10 days can be simulated at flow tube residence times of a few minutes or less. Recent experimental studies suggest that flow reactor-generated SOA particles have compositions similar to SOA generated in smog chambers (Bruns et al., 2015; Lambe et al., 2015) and in the atmosphere (Tkacik et al., 2014; Ortega et al., 2016; Palm et al., 2016). Modeling studies suggest that flow reactors can simulate tropospheric oxidation reactions with minimal experimental artifacts (Li et al., 2015; Peng et al., 2015, 2016). A limitation of flow reactors is the need

to use parts-per-million levels of $O_3$, hindering the possibility to efficiently simulate $NO_x$-dependent SOA formation pathways. Simple addition of NO to flow reactors, while possible (Liu et al., 2015), cannot sustain $NO_x$ mixing ratios at levels that are sufficient to compete with hydroperoxy ($HO_2$) radicals as a sink for organic peroxy ($RO_2$) radicals due to fast conversion of $NO_x$ to nitric acid ($HNO_3$) via the reactions $NO + O_3 \rightarrow NO_2$ and $NO_2 + OH \rightarrow HNO_3$. Here, we present a new method
well suited to the characterization of $NO_x$-dependent SOA formation pathways in oxidation flow reactors. By utilizing $O(^1D)$ radicals that are generated from $O_3$ photolysis, we add $N_2O$ to generate NO via the reaction $O(^1D) + N_2O \rightarrow 2NO$ with no additional method modifications. We validate the concept using high-resolution time-of-flight chemical ionization mass spectrometer measurements (HR-ToF-CIMS) to detect gas-phase oxidation products of isoprene and $\alpha$-pinene that have been observed in $NO_x$-influenced environments and laboratory chamber experiments.

## 2  Experimental

Experiments were conducted using an Aerodyne Potential Aerosol Mass (PAM) oxidation flow reactor, which is a horizontal 13.3 L aluminum cylindrical chamber (46 cm long $\times$ 22 cm ID) operated in continuous flow mode (Kang et al., 2007; Lambe et al., 2011a). The average residence time was 80 s. The relative humidity (RH) in the reactor was controlled in the range of 3–35 % at 22°C, corresponding to $H_2O$ mixing ratios of approximately 0.07 - 1%. The irradiance in the reactor was
measured using a photodiode (TOCON-C6, sglux Gmbh). The gas-phase SOA precursors used in these studies include two biogenic compounds (isoprene, $\alpha$-pinene) that were prepared in compressed gas cylinders and introduced to the reactor at controlled rates using a mass-flow controller. Mixing ratios of the gas-phase precursors entering the reactor were 36 ppb for isoprene (diluted from 1000 ppm in $N_2$, Matheson) and 15 ppb for $\alpha$-pinene (diluted from 150 ppm in $N_2$, Matheson). These mixing ratios are a factor of 3 to 10 lower than mixing ratios that are typically required to induce homogenous nucleation of
condensable oxidation products in related oxidation flow reactor studies (Lambe et al., 2011b). Minimizing precursor mixing ratios also decreases the rate of $RO_2$ self-reactions relative to $RO_2 + HO_2$ and $RO_2 + NO$ reactions. This is a goal for most laboratory experiments that is not specific to the method proposed here. However, this goal takes on added importance when $RO_2$ can be formed via OH, $O_3$ and/or $NO_3$ oxidation using this method as discussed in Section 2.1.

### 2.1  OH radical and $NO_x$ generation

OH radicals were produced in the reactor via the reaction $O(^1D) + H_2O \rightarrow 2OH$, with $O(^1D)$ radicals produced from the reaction $O_3 + h\nu \rightarrow O_3 + O(^1D)$. $O_3$ ($\sim$1-5 ppm) was generated outside the flow reactor by $O_2$ irradiation at 185 nm using a mercury fluorescent lamp (GPH212T5VH, Light Sources, Inc.). $O(^1D)$ was produced by photolysis of $O_3$ at 254 nm inside the reactor using two or four mercury fluorescent lamps (GPH436T5L, Light Sources, Inc). A fluorescent dimming ballast was used to regulate current applied to the lamps. To vary [OH] inside the reactor, $I_{254}$ was varied by changing the dimming voltage
applied to the ballast between 1.6 and 10 VDC. At these conditions, $I_{254}$ ranged from approximately $(0.064 - 3.2) \times 10^{15}$ ph $cm^{-2}$ sec. The highest $I_{254}$ value was calculated from the internal surface area of the reactor and the lamp output at maximum

intensity (e.g. 10 VDC) specified by the manufacturer. Lower $I_{254}$ values were calculated from the measured irradiance at lower dimming voltage relative to the measured irradiance and manufacturer-specified lamp output at 10 VDC.

NO and $NO_2$ were produced via the reaction $N_2O + O(^1D) \rightarrow 2NO$, followed by the reaction $NO + O_3 \rightarrow NO_2$. $N_2O$ (99.5%) was introduced from a compressed gas cylinder at flow rates ranging from 0 to 648 $cm^3$ $min^{-1}$, corresponding to mixing ratios of 0% to 5.6% at the carrier gas flow rates that were used. Using $N_2O$ as the $NO_x$ precursor has the following advantages over the simple addition of NO to the carrier gas. First, due to continuous production of $O(^1D)$ from $O_3$ photolysis inside the reactor (along with minor consumption of $N_2O$), the spatial distribution of NO and $NO_2$ is more homogenous. Second, attainable steady-state mixing ratios of NO from $N_2O + O(^1D)$ reactions (ppb levels) are orders of magnitude higher than simple NO injection (sub-ppt levels) as inferred from photochemical model simulations described below in Sect. 2.3. Third, photolysis of $N_2O$ at 185 nm (if used) provides an additional source of $O(^1D)$ from the reaction $N_2O + h\nu \rightarrow N_2 + O(^1D)$. We assume background [NO] < 0.05 ppb in the reactor based on separate [NO] measurements and calculate additional NO formed from $N_2O + O(^1D)$ reactions using the model described in Sect. 2.3. Gradients in $[O(^1D)]$ due to its reaction with $H_2O$ and $N_2O$ may alter spatial distributions of $O_x$, $HO_x$ and $NO_x$ in the reactor. To first order, gradients in $[O(^1D)]$ decrease both $[HO_2]$ and [NO] to a similar extent, keeping the relative rates of $RO_2 + HO_2$ and $RO_2 + NO$ termination pathways the same.

In most cases, oxidation of VOCs by $O_3$ is slower than oxidation by OH radical, even with parts per million levels of $O_3$ present (Peng et al., 2016). $NO_3$ radicals, which are produced as a byproduct of $NO_2 + O_3$ or $HNO_3 + OH$ reactions, can potentially convolute interpretation of results if the relative oxidation rates of isoprene/$\alpha$-pinene by OH and $NO_3$ are comparable. For results presented in Sects. 3.3 and 3.4, calculated OH, $O_3$ and $NO_3$ exposures combined with published OH, $O_3$ and $NO_3$ rate constants (Atkinson, 1986, 1991; Grosjean and Grosjean, 1996) suggest that the relative contribution of $NO_3$ to isoprene and $\alpha$-pinene oxidation ranges from approximately 0 to 4% and 0 to 60%, respectively, as a function of $[N_2O]$. Thus, reaction rates of $\alpha$-pinene with OH, $O_3$ and $NO_3$ may be comparable under a subset of experimental conditions. Potential implications are discussed in more detail in Sections 3.3.4 and 3.4.4.

## 2.2 $NO_x$ and chemical ionization mass spectrometer (CIMS) measurements

In one set of experiments, [NO] and $[NO_2]$ were measured downstream of the reactor with a Thermo Scientific Model 42i chemiluminescent analyzer and an Aerodyne Cavity Attenuated Phase Shift (CAPS) $NO_2$ analyzer, which measures $NO_2$ absorption at $\lambda$ = 450 nm (Kebabian et al., 2008). During these experiments, the following operating conditions were used: $I_{254}$ = $4 \times 10^{15}$ ph $cm^{-2}$ $sec^{-1}$, $[O_3]$ = 1 ppm, $[H_2O]$ = 0.07% and 1%, $[N_2O]$ = 0 to 3%. These conditions assess a subset of the attainable operating conditions for comparison with outputs of the photochemical model described in Section 2.3. The measured $NO_2$ mixing ratio was decreased by 10 ppb due to absorption by 1 ppm $O_3$ at 450 nm in the absence of $NO_2$. The measured NO mixing ratio was scaled by a factor of 3.2 for depletion downstream of the reactor due to 1.2 sec reaction time with 1 ppm $O_3$ in the sample line, assuming $k_{O_3}^{NO}$ = $1.8 \times 10^{-14}$ $cm^3$ $molec^{-1}$ $sec^{-1}$ and pseudo-first order conditions (Atkinson et al., 2004). Additional NO depletion inside the chemiluminscent analyzer ($\sim$ 47% at 1 ppm $O_3$) was accounted for in a separate experiment where known mixing ratios of NO (50 ppb) and $O_3$ (0 to 6.9 ppm) were added at the inlet of the

instrument (Fig. S1). Because the combined NO depletion in the sample line and the chemiluminscent analyzer is significantly higher at higher [$O_3$] (e.g. $\sim$90% at [$O_3$] = 2 ppm and $\sim$99.6% at [$O_3$] = 5 ppm), accurate experimental characterization of [NO] is more difficult above [$O_3$] $\sim$ 1 ppm.

In another set of experiments, mass spectra of isoprene and $\alpha$-pinene gas-phase oxidation products were obtained with an Aerodyne high-resolution time-of-flight mass spectrometer (Bertram et al., 2011) coupled to an atmospheric pressure interface with a nitrate ion chemical ionization source ($NO_3^-$-HRToF-CIMS, hereafter abbreviated as "$NO_3^-$-CIMS") (Eisele and Tanner, 1993; Ehn et al., 2012). Nitrate ($NO_3^-$) and its higher order clusters (e.g. $HNO_3NO_3$) generated from x-ray ionization of $HNO_3$ were used as the reagent due to the selectivity to highly oxidized organic compounds, including species that contribute to SOA formation (Ehn et al., 2014; Krechmer et al., 2015). Isoprene and $\alpha$-pinene oxidation products were detected as adducts with $NO_3^-$ or $HNO_3NO_3^-$. CIMS data were analyzed using the Tofware software package (Tofwerk AG, Aerodyne Research, Inc.) implemented in IGOR Pro 6 (Wavemetrics, Inc.). The output of the PAM oxidation flow reactor was sampled at 10.5 $\mathrm{Lmin^{-1}}$ through a 2' length of 0.75" OD stainless steel tubing inserted directly into the rear feedthrough plate of the reactor.

Ambient $NO_3^-$-CIMS measurements were conducted during the Southern Oxidant and Aerosol Study (SOAS) at the forest site in Centreville, AL (June 1 - July 15, 2013). At this site, emissions were dominated by local biogenic volatile organic compounds (BVOC) with occasional influence from nearby anthropogenic sources (Hansen et al., 2003). The mixing of biogenic and anthropogenic emissions at the foreest site promotes the formation of organic nitrates via oxidation of BVOC in the presence of $NO_x$ (Lee et al., 2016).

## 2.3 Photochemical modeling

We used a photochemical model (Li et al., 2015; Peng et al., 2015) implemented in MATLAB (Mathworks) to calculate concentrations of radical/oxidant species produced in the reactor. Model input parameters included pressure, temperature, [$H_2O$], [$O_3$], [$N_2O$], $I_{254}$, mean residence time, and the input mixing ratios of isoprene and $\alpha$-pinene. Differential equations used to describe the radical/oxidant chemistry were integrated at 5 millisecond time steps. The following reactions and associated kinetic rate constants (Sander et al., 2000, 2006) were implemented to describe $NO_x$ chemistry in the reactor:

$$N_2O + h\nu \rightarrow N_2 + O(^1D) \tag{R1}$$

$$N_2O + O(^1D) \rightarrow 2NO \tag{R2}$$

$$N_2O + O(^1D) \rightarrow N_2 + O_2 \tag{R3}$$

$$NO + OH + M \rightarrow HONO + M \tag{R4}$$

$$NO + HO_2 \rightarrow OH + NO_2 \tag{R5}$$

$$NO + O_3 \rightarrow NO_2 + O_2 \tag{R6}$$

$$NO_2 + O \rightarrow NO + O_2 \tag{R7}$$

$$NO_2 + O_3 \rightarrow NO_3 + O_2 \tag{R8}$$

$$NO_3 + O \rightarrow NO_2 + O_2 \tag{R9}$$

$$HONO + OH \rightarrow H_2O + NO_2 \tag{R10}$$

$$NO_2 + NO_3 + M \rightleftharpoons N_2O_5 + M \tag{R11}$$

$$NO + NO_3 \rightarrow 2NO_2 + O_2 \tag{R12}$$

$$NO_2 + OH + M \rightarrow HNO_3 + M \tag{R13}$$

$$NO_2 + HO_2 + M \rightarrow HNO_4 + M \tag{R14}$$

$$N_2O_5 + H_2O \rightarrow 2HNO_3 \tag{R15}$$

$$HNO_3 + OH \rightarrow H_2O + NO_3 \tag{R16}$$

$$NO_3 + NO_3 \rightarrow 2NO_2 + O_2 \tag{R17}$$

The model also includes simplified $RO_2$ chemistry, which is incorporated using the reactions listed below (IUPAC, 2013). The addition of these reactions constrain the effects of added isoprene or $\alpha$-pinene (species "X" below) on steady-state [OH], [HO$_2$] and [NO]. Second-generation organic radical products of initial organic radical reactions ("RPHO", "$RPO_2$", "RPO") are not reacted further in the model.

$$\text{OH} + \text{X} \rightarrow \text{RO}_2 + \text{H}_2\text{O} \tag{R18}$$

$$\text{RO}_2 + \text{NO} \rightarrow \text{RO} + \text{NO}_2 \tag{R19}$$

$$\text{RO}_2 + \text{HO}_2 \rightarrow \text{ROOH} + \text{O}_2 \tag{R20}$$

$$\text{ROOH} + \text{OH} \rightarrow \text{RO}_2 + \text{H}_2\text{O} \tag{R21}$$

$$\text{ROOH} + \text{OH} \rightarrow \text{RPHO} + \text{OH} + \text{H}_2\text{O} \tag{R22}$$

$$\text{RO}_2 + \text{OH} \rightarrow \text{RPO}_2 + \text{H}_2\text{O} \tag{R23}$$

$$\text{RO}_2 + \text{RO}_2 \rightarrow \text{ROOR} \tag{R24}$$

$$\text{RO} + \text{O}_2 \rightarrow \text{RPO} + \text{HO}_2 \tag{R25}$$

$$\text{RO}_2 + \text{NO} + \text{M} \rightarrow \text{RO}_2\text{NO} + \text{M} \tag{R26}$$

$$\text{RO} + \text{NO} + \text{M} \rightarrow \text{RONO} + \text{M} \tag{R27}$$

$$\text{RO} + \text{NO}_2 + \text{M} \rightarrow \text{RONO}_2 + \text{M} \tag{R28}$$

Calculated OH exposures (product of mean OH concentration and residence time) ranged from $1.7 \times 10^{10}$ to $2.1 \times 10^{12}$ molec cm$^{-3}$ sec or approximately 3 hours to 16 days of equivalent atmospheric exposure at [OH] = $1.5 \times 10^6$ cm$^{-3}$ (Mao et al., 2009). Steady-state [NO] and [HO$_2$] ranged from 0 to 13.5 ppb and 0.01 to 2.1 ppb, respectively, depending on [N$_2$O], [H$_2$O], [O$_3$] and I$_{254}$. We assumed $\pm 25\%$ uncertainty in the calculated OH exposure and $\pm 60\%$ uncertainty in other model outputs (Peng et al., 2015). For ratios of model outputs with independent $\pm 60\%$ uncertainties (e.g. NO:HO$_2$), propagated uncertainties of $\pm 85\%$ were assumed. Addition of N$_2$O at the highest mixing ratios that were used suppressed [OH] because N$_2$O competes with H$_2$O as a sink for O($^1$D). Potential consequences of OH suppression are discussed where applicable in Sects. 3.3 and 3.4.

## 3 Results and Discussion

### 3.1 Comparison of measured and modeled [NO] and [NO$_2$] values following O($^1$D) + N$_2$O and NO + O$_3$ reactions

Figure 1 compares modeled and measured NO mixing ratios obtained following 80 sec residence time in the reactor at the operating conditions described in Sect.2.2. The corresponding integrated OH exposures are approximately $2.6 \times 10^{11}$ and $2.4 \times 10^{12}$ molec cm$^{-3}$ sec, respectively, in the absence of added N$_2$O. Symbols are colored by [N$_2$O] which ranged from 0 to 3%. Measured [NO] ranged from 0 to 10.4 ppb and increased with increasing [N$_2$O], as expected, at both [H$_2$O] = 0.07% and 1%. The mean ratio of modeled-measured [NO] was $0.94 \pm 0.19$ at [H$_2$O] = 0.07% and $3.85 \pm 2.33$ at [H$_2$O] = 1%.

NO$_2$, which is formed by the NO + O$_3$ reaction, is more straightforward to measure under these conditions because NO$_2$ reacts approximately 500 times slower than NO with O$_3$. Thus, a comparison of modeled and measured [NO$_2$] provides additional method evaluation with less uncertainty than [NO] measurements. Figure 2 compares corresponding modeled and measured NO$_2$ mixing ratios obtained during the same experiments described in Figure 1. As expected, [NO$_2$] increased

with increasing $[N_2O]$ because of faster $NO + O_3$ reaction rate from increasing $[NO]$. At $[H_2O] = 0.07\%$, measured $[NO_2]$ ranged from 0 to 291 ppb, whereas at $[H_2O] = 1\%$, measured $[NO_2]$ ranged from 0 to 59 ppb. $[NO_2]$ was lower in the latter case because additional OH was formed from $O(^1D) + H_2O$ reactions (Section 2.1), which increased the rate of the OH + $NO_2$ reaction. The mean ratio of modeled-measured $[NO_2]$ was $0.72 \pm 0.39$ at $[H_2O] = 0.07\%$ and $1.05 \pm 0.50$ at $[H_2O] = 1\%$. These results, combined with results shown in Figure 1, suggest that an uncharacterized $H_2O$- or $HNO_3$-related artifact negatively biased the measured $[NO]$ values at $[H_2O] = 1\%$, and that the photochemical model described in Section 2.3 may be used to evaluate a wider range of reactor operating conditions. The model also constrains mixing ratios of radical species such as $HO_2$ that are difficult to measure directly or require additional measurement techniques (Mauldin et al., 1999; Sanchez et al., 2016).

## 3.2 Optimal reactor operating conditions for $O(^1D) + N_2O +$ reactions

To investigate optimal operating conditions for $NO_x$ generation, we implemented the model described in Sect. 2.3 over operating conditions $I_{254} = 3.2 \times 10^{13}$ to $6.4 \times 10^{15}$ ph cm$^{-2}$ sec$^{-1}$, $[O_3] = 0.5$ to 50 ppm, and $[H_2O] = 0.07$ to $2.3\%$ at $22°C$, respectively, as a function of $[N_2O] = 0$ to $5\%$. These values span the nominal range of operating conditions that can be achieved with the PAM reactor. To facilitate independent evaluation of the effects of $[O_3]$ and $I_{254}$ on $[NO]$, we restricted our analysis to conditions that use only 254 nm photolysis. Using both 185 and 254 nm photolysis provides additional sources of $O(^1D)$ and OH from $N_2O$ and $H_2O$ photolysis at 185 nm, respectively, at the expense of independent control of $[O_3]$ and $I_{254}$.

Figure 3 shows the modeled steady-state $[NO]$ in the reactor as a function of $[N_2O] = 0$ to $5\%$, assuming a mean residence time of 80 sec, $[H_2O] = 1\%$, and $[O_3] = 5$ ppm. In addition, Figs. S1 - S3 in the Supplement show modeled $NO:HO_2$ and $OH:NO_3$ ratios as a function of input $[N_2O]$, with $I_{254}$, $[O_3]$, and $[H_2O]$ each varied individually while other input conditions are fixed. The following observations that are obtained from Figs. 3 and S1 - S3 were used to identify the optimal operating conditions:

1. At fixed $[O_3]$, $[H_2O]$, and $[N_2O]$, $[O(^1D]$ and $[NO]$ increase with increasing $I_{254}$ (Figs. 3 and S1).

2. At fixed $I_{254}$, $[H_2O]$, and $[N_2O]$, increasing $O_3$ increases the production and loss rates of NO from $N_2O + O(^1D)$ and $NO + O_3$ reactions, respectively. The relative importance of NO + OH, $NO + O_3$, and $NO + NO_3$ reactions, which depend on $[N_2O]$ and $[O_3]$, further influence $[NO]$:

   – At $[N_2O] \sim 1\%$, increasing $[O_3]$ from 0.5 to 5 ppm increases $[NO]$ because the reaction rate of NO + OH decreases relative to $NO + O_3$ (Fig. S2a).

   – At $[N_2O] > 1\%$, increasing $[O_3]$ from 5 to 50 ppm decreases $[NO]$ because the reaction rate of $NO + NO_3$ increases relative to $NO + O_3$ (Fig. S2a).

3. At fixed $I_{254}$, $[H_2O]$, and $[N_2O]$, increasing $[O_3]$ decreases $[NO]:[HO_2]$ and $[OH]:[NO_3]$ by increasing $NO_2$ and $NO_3$ formation from $NO + O_3$ and $NO_2 + O_3$ reactions.

4. At fixed $I_{254}$, $[O_3]$, and $[N_2O]$, increasing $[H_2O]$ increases $[OH]:[NO_3]$ by increasing OH production from $H_2O$ + $O(^1D)$ reactions (Fig. S3).

The relative importance of these operating conditions is situationally dependent on the relative OH, $O_3$, and $NO_3$ rate constants of the target species and photochemical age. To demonstrate proof of principle, we present $NO_3^-$-CIMS spectra of isoprene and $\alpha$-pinene oxidation products in the following sections.

### 3.3 $NO_3^-$-CIMS spectra of isoprene oxidation products

Figure 4 shows $NO_3^-$-CIMS mass spectra of products generated from the oxidation of isoprene ($C_5H_8$) that cluster with $NO_3^-$ ions to form $NO_3^-$-species adducts. Ion signals are plotted as a function of mass-to-charge ratio (m/Q). $NO_3$ adduct formation is a relatively low-energy process that does not result in fragmentation of the analyte (Eisele and Tanner, 1993; Kurtén et al., 2011). Thus, the measured ion signals are directly related to the chemical formulas of individual species that are generated in the reactor. Ion signals corresponding to isoprene oxidation products shown in Fig. 4 were colored based on classification in ion groups containing 2-5 carbon atoms with zero ($C_4H_{4,6,8}O_{4-7}$ and $C_5H_{6,8,10,12}O_{3-8}$), one ($C_{2-3}H_{3,5}NO_5$ and $C_5H_{7,9,11}NO_{6-11}$), and two ($C_5H_{10}N_2O_{8-10}$) nitrogen atoms, where we assumed that nitrogen atoms were associated with nitrate functional groups and not heterocyclic compounds. We also assume that nitrate functional groups are formed from $RO_2$ + NO or $RO_2$ + $NO_2$ reactions (Sect. 2.1). To examine changes in relative contributions of $C_4H_{4,6,8}O_{4-7}$, $C_5H_{6,8,10,12}O_{3-8}$, $C_5H_{7,9,11}NO_{6-11}$, and $C_5H_{10}N_2O_{8-10}$ ions as a function of added $NO_x$, we made two simplifying assumptions: (1) the $NO_3^-$-CIMS had the same sensitivity to all species that were detected, and (2) $HNO_3$ generated in the reactor did not alter the relative selectivity of the CIMS to different classes of oxidation products, as has been observed in some cases (Hyttinen et al., 2015).

To generate spectra shown in Fig. 4, the reactor was operated at $I_{254}$ = $6.4 \times 10^{13}$ and $3.2 \times 10^{15}$ ph cm$^{-2}$ sec$^{-1}$, $[H_2O]$ = 1%, and $[N_2O]$ = 0 and 3%. As shown in Figs. S4 and S5, corresponding OH exposures ranged from $(1.7 - 2.0) \times 10^{10}$ (Fig. 4a and 4c; calculated > 82% of isoprene reacted) and $(0.52 - 2.1) \times 10^{12}$ molec cm$^{-3}$ sec (Fig. 4b and 4d; calculated $\sim$ 100% of isoprene reacted), respectively. At low OH exposure, the OH suppression at "high $NO_x$" relative to "low $NO_x$" was comparatively minor (15%), whereas at high OH exposure, the OH suppression at "high $NO_x$" relative to 'low $NO_x$" was larger (75%). At the "high-$NO_x$" OH exposure of $5.2 \times 10^{11}$ molec cm$^{-3}$ sec, isoprene can react with OH up to 52 times in the reactor. This presumably exceeds the number of OH reactions (followed by $RO_2$ + NO reactions) that are necessary to fragment or condense oxidation products to the point where they are no longer detected with $NO_3^-$-CIMS. Thus, it is unlikely that OH suppression at "high OH" and "high $NO_x$" significantly affected the $NO_3^-$-CIMS spectra shown in Fig. 4. To aid interpretation of results shown in Fig. 4, Fig. 5 summarizes several known isoprene + OH reaction pathways that are terminated by reactions of $RO_2$ with $HO_2$, NO, or $NO_2$. As will be discussed in the following sections, these pathways yield multigenerational oxidation products with chemical formulas corresponding to the major ions that are plotted in Fig. 4.

### 3.3.1 $NO_3^-$-CIMS spectral features observed at "low $NO_x$" conditions

$C_{4-5}H_{4-12}O_{3-8}$ ions comprised 93% and 97% of the signals at low and high OH exposure (Figs. 4a and Fig. 4c, respectively). The $C_5H_{7-11}NO_{6-11}$ signals that were observed here may be due to background $NO_x$ in the reactor (Sect. 2.1). The signal at m/Q = 230, $C_5H_{12}O_6$ ($NO_3^-$ omitted for brevity here and elsewhere), was the largest signal detected at both low and high OH exposures at "low-$NO_x$" conditions. Figure 5 suggests this species is a second-generation oxidation product generated from two reactions with OH and two $RO_2$ + $HO_2$ termination reactions (Krechmer et al., 2015; St. Clair et al., 2016) and is typically associated with isoprene SOA formation and growth under "low-$NO_x$" conditions. (Liu et al., 2016) Signals in Figs. 4b and 4d are approximately 10 times higher than in Figs. 4a and 4c because additional OH exposure produces higher yields of multi-generation oxidation products that are detected with $NO_3^-$-CIMS.

Previously-identified multi-generation isoprene oxidation products such as $C_5H_{10}O_5$, $C_5H_{12}O_5$, and $C_5H_{10}O_6$ (Surratt et al., 2006; Krechmer et al., 2015; St. Clair et al., 2016) were also detected at significant intensity under low-$NO_x$ conditions. These species are formed after two reactions with OH, one $RO_2$+ $HO_2$ termination reaction and one $RO_2$ + $RO_2$ termination reaction (Fig. 5). When the OH exposure was increased from $2.0 \times 10^{10}$ to $2.1 \times 10^{12}$ molec cm$^{-3}$ sec, the signal at $C_5H_{12}O_6$ increased by a factor of 10 and the signal at m/Q = 246, $C_5H_{12}O_7$, increased by a factor of 5. At high OH exposure, $C_5H_{12}O_7$ was the second-largest peak in the spectrum. These highly oxygenated isoprene oxidation products are likely also important in SOA formation processes. $C_5H_{10}O_7$ is a proposed tri-hydroperoxy carbonyl product formed after one reaction with OH, two hydrogen shifts and one $RO_2$ + $HO_2$ termination reaction as shown in Fig. 5 (Peeters et al., 2014).

We hypothesize two reasons for the prominence of $C_5H_{10}O_7$, $C_5H_{12}O_7$, and $C_5H_{10}O_8$ in our spectra. First, $NO_3^-$ is more selective to highly oxidized species than other reagent ions (Surratt et al., 2006; Liu et al., 2016). Second, higher OH exposures were achieved in the reactor than in environmental chambers. For example, the spectra shown in Figs. 4a and 4b were obtained at integrated OH exposures of $1.7 \times 10^{10}$ and $2.1 \times 10^{12}$ molec cm$^{-3}$ sec, respectively, compared to an OH exposure of $8.6 \times 10^9$ molec cm$^{-3}$ sec in the environmental chamber $NO_3^-$-CIMS measurements conducted by Krechmer et al. (2015).

### 3.3.2 $NO_3^-$-CIMS spectral features observed at "high $NO_x$" conditions

Following addition of $N_2O$ at ~3% mixing ratio, the $NO_3^-$-CIMS spectra changed significantly at low and high OH exposures (Figs. 4b and 4d). The signals of $C_{4-5}H_{4-12}O_{3-8}$ oxidation products decreased, although the $C_4H_{4,6,8}O_{4-7}$ : $C_5H_{6,8,10,12}O_{3-8}$ ratio increased, presumably due to decomposition of alkoxy (RO) radicals generated from $C_5$ $RO_2$ + NO reactions into $C_4$ products. The $C_{2-3}H_{3,5}NO_5$ (peroxy acetyl nitrate and peroxy propionyl nitrate), $C_5H_{7,9,11}NO_{6-11}$ and $C_5H_{10}N_2O_{8-10}$ signals increased. At low OH exposure, $C_{2-3}H_{3,5}NO_5$, $C_5H_{7,9,11}NO_{6-11}$ and $C_5H_{10}N_2O_{8-10}$ signals constituted 2%, 38% and 7% of the $NO_3^-$-CIMS signals, respectively (Fig. 4c). The largest signal in this spectrum was m/Q = 259, $C_5H_{11}NO_7$. Fig. 5 suggests this compound is a second-generation oxidation product that is formed after two reactions with OH, one $RO_2$ + NO termination reaction and one $RO_2$ + $HO_2$ termination reaction (Xiong et al., 2015). The signal observed at m/Q = 288, $C_5H_{10}N_2O_8$, is a second-generation oxidation product that is formed after two reactions with OH and two $RO_2$ + NO termination reactions (Fig. 5) (Xiong et al., 2015). Other ion signals associated with dinitrate species included m/Q =

304, $C_5H_{10}N_2O_9$, and m/Q = 320, $C_5H_{10}N_2O_{10}$. Related signals were detected at m/Q = 351 and 367 (not shown), which we assume represent $(HNO_3NO_3^-)C_5H_{10}N_2O_8$ and $(HNO_3NO_3^-)C_5H_{10}N_2O_9$ because we are not aware of other feasible $(NO_3^-)C_5$ adducts at these mass-to-charge ratios.

At high OH exposure, the same $C_5H_{7,9,11}NO_{6-11}$ and $C_5H_{10}N_2O_{8-10}$ species observed at low OH exposure were detected, but at higher concentrations and at higher dinitrate:nitrate. This is presumably due to higher NO:$HO_2$ achieved at higher $I_{254}$ and fixed $[N_2O]$ (Figs. 3, S2, S5-S6). $C_{2-3}H_{3,5}NO_5$, $C_5H_{7,9,11}NO_{6-11}$ and $C_5H_{10}N_2O_{8-10}$ signals made up 0.3%, 33% and 56%, respectively, of the $NO_3^-$-CIMS spectrum shown in Fig. 4d, where $C_5H_{10}N_2O_8$ was the largest signal that is detected.

To demonstrate our ability to mimic atmospheric $NO_x$-dependent photochemistry, Figures 4e and 4f show $C_4H_{4,6,8}O_{4-7}$, $C_5H_{6,8,10,12}O_{3-8}$, $C_{2-3}H_{3,5}NO_5$, $C_5H_{7,9,11}NO_{6-11}$, and $C_5H_{10}N_2O_{8-10}$ ion signals detected in $NO_3^-$-CIMS spectra at the SOAS ground site in Centreville, Alabama, USA. The spectra shown were obtained on 25 Jun 2013 (0730-1100) and 4-5 Jul 2013 (1200 – 0000) which represented periods with sustained "high" and "low" NO mixing ratios of $0.53 \pm 0.17$ ppb and $0.024 \pm 0.025$ ppb, respectively, measured at the site. Figures 4a, 4c and 4e indicate that adding $N_2O$ to the reactor increases the similarity between the composition of isoprene oxidation products generated at lower photochemical age in the reactor (Figs. 4a and 4c) and under "low-NO" ambient conditions (Fig. 4e). Likewise, Figures 4b, 4d and 4f indicate that adding $N_2O$ to the reactor increases the similarity between the composition of isoprene oxidation products generated at higher photochemical age in the reactor (Figs. 4b and 4d) and at "high-NO" ambient conditions (Fig. 4f). $(HNO_3NO_3^-)C_5H_{10}N_2O_{8-9}$ adducts were also observed in Fig. 4f (not shown).

### 3.3.3 Influence of acylperoxy nitrates from $RO_2 + NO_2$ reactions

Acylperoxy nitrates (APNs) may be generated from reactions of aldehydic, biogenic VOC oxidation products with OH followed by $RO_2 + NO_2$ termination reactions, e.g. LaFranchi et al. (2009). Peroxy acetyl nitrate (PAN, $C_2H_3NO_5$) and propionyl peroxy nitrate (PPN, $C_3H_5NO_5$), are minor components (<2%) of the spectra shown in Figs. 4c - 4f. A comparison of Figs. 4c and 4e suggests that yields of PAN and PPN are not enhanced in the reactor compared to atmospheric conditions. Additional APNs may be generated following the OH oxidation of methacrolein, a first-generation isoprene oxidation product. Methacryloyl peroxy nitrate (MPAN, $C_4H_5NO_5$) is a second-generation oxidation product formed after one methacrolein + OH reaction and one $RO_2 + NO_2$ termination reaction (Orlando et al., 1999). C4-hydroxynitrate-PAN ($C_4H_6N_2O_9$) is a third-generation oxidation product formed through the methacrolein channel after three reactions with OH, two $RO_2 + NO$ termination reactions and one $RO_2 + NO_2$ termination reaction (Surratt et al., 2010). Neither $C_4H_5NO_5$ nor $C_4H_6N_2O_9$ were detected in the laboratory and ambient $NO_3^-$-CIMS spectra shown in Figs. 4c - 4f. Either these compounds were oxidized or thermally decomposed prior to detection, or their signals were below detection limit. $C_4H_7NO_5$, which is formed after one methacrolein + OH reaction and one $RO_2 + NO$ termination reaction (Surratt et al., 2010), was detected (Fig. 5). Taken together, these observations suggest that yields of APNs were not significantly enhanced in the reactor compared to atmospheric conditions.

### 3.3.4 Influence of isoprene + NO₃ reactions

Based on the calculated isoprene + OH and isoprene + $NO_3$ reaction rates (Figs. S5-S6) we assume that isoprene + $NO_3$ reactions were a minor influence on the $NO_3^-$-CIMS spectra shown in Figs. 4c and 4d. This assumption is further supported by the similarity between laboratory and ambient $NO_3^-$-CIMS spectra, the latter of which was obtained during the daytime and thus with minimal $NO_3$ exposure (0730 - 1100 for "high-NO" spectra shown in Fig. 4f). Specific operating conditions different than those used in this study could increase the relative influence of isoprene + $NO_3$ reactions. In this hypothetical situation, enhanced yields of $C_5H_7NO_5$, $C_5H_8N_2O_8$ and $C_5H_{10}N_2O_8$ might occur following two reactions with $NO_3$ (Rollins et al., 2009). In addition, $C_5H_{10}N_2O_9$ may be generated from one isoprene + $NO_3$ reaction followed by one $RO_2$ + $HO_2$ termination reaction (Schwantes et al., 2015). All four of these ions are detected in the spectra shown in Fig. 4, although $C_5H_8N_2O_8$ (not shown in Fig. 4) is present at 0.5% of the intensity of $C_5H_{10}N_2O_8$. If $C_5H_8N_2O_8$ : $C_5H_{10}N_2O_8$ is significantly different under $NO_3$-dominated conditions, this ratio could distinguish the relative rates of isoprene + OH and isoprene + $NO_3$ reactions. Otherwise, it is not clear that the expected product distributions are significantly different whether isoprene is oxidized by OH or $NO_3$ in the presence of $NO_x$.

### 3.4 $NO_3^-$-CIMS spectra of $\alpha$-pinene oxidation products

Figure 6 shows $NO_3^-$-CIMS mass spectra of products generated from the oxidation of $\alpha$-pinene ($C_{10}H_{16}$). Ion signals corresponding to $\alpha$-pinene oxidation products were colored based on classification in $C_5H_{6,8}O_{5-7}$, $C_{6-9}H_{8,10,12,14}O_{6-12}$, $C_{10}H_{14,16,18}O_{5-14}$, and $C_{19-20}H_{28,30,32}O_{9-18}$ ion groups containing zero nitrogen atoms; $C_{2-3}H_{3,5}NO_5$, $C_5H_7NO_{6-11}$, $C_{6-9}H_{9,11,13,15}NO_{5-10}$, and $C_{10}H_{15,17}NO_{4-14}$ ion groups containing one nitrogen atom; and a $C_{10}H_{16,18}N_2O_{6-13}$ ion group containing two nitrogen atoms. As was the case with isoprene oxidation products, we assumed nitrogen atoms present in $\alpha$-pinene oxidation products were associated with nitrate functional groups formed from $RO_2$ + NO or $RO_2$ + $NO_2$ reactions. Additionally, we again assumed the $NO_3^-$-CIMS had the same sensitivity to all species that were detected, and that $HNO_3$ generated in the reactor did not alter the relative selectivity of the CIMS to different classes of oxidation products. Because the oxidation pathways leading to $\alpha$-pinene-derived HOM are signfcantly more complex than those leading to isoprene-derived HOM, the analogous figure to Fig. 5 for $\alpha$-pinene-derived HOM is beyond the scope of this paper.

To generate spectra shown in Fig. 6, the reactor was operated at $I_{254} = 2.8 \times 10^{15}$ ph $cm^{-2}sec^{-1}$, $[H_2O] = 0.07\%$, and $[N_2O] = 0$ and 3.2%. In this experiment, lower $[H_2O]$ was used to minimize [OH] and facilitate closer comparison with spectra from previous $NO_3^-$-CIMS studies of $\alpha$-pinene + $O_3$ oxidation products generated at "low-$NO_x$" conditions (Ehn et al., 2012, 2014). As shown in Fig. S7, corresponding OH and $O_3$ exposures ranged from $(0.19 - 1.8) \times 10^{11}$ molec $cm^{-3}$ sec and $(7.2-9.5) \times 10^{16}$ molec $cm^{-3}$ sec for the low- and high-$NO_x$ conditions, respectively. To first order, at OH and $O_3$ exposures of $2.1 \times 10^{10}$ and $7.4 \times 10^{15}$ molec $cm^{-3}$ sec that are attained at $[N_2O] = 3.2\%$, $\alpha$-pinene should react once with each oxidant in the gas phase. Thus, at the highest $[N_2O]$ used, yields of second-generation (or later) $\alpha$-pinene + OH oxidation products detected with the $NO_3^-$-CIMS were minimized relative to $\alpha$-pinene + $O_3$ first-generation oxidation products, as desired (Jokinen et al., 2015). However, a potential consequence of using $O(^1D)$ + $N_2O$ reactions to study the $NO_x$-dependence of chemical systems similar

to those examined by Ehn et al. (2012, 2014) is that $RO_2$ may be produced from $\alpha$-pinene + $NO_3$ reactions in addition to $\alpha$-pinene + $O_3$ or $\alpha$-pinene + OH reactions (Sect. 2.1 and Fig. S7).

### 3.4.1  $NO_3^-$-CIMS mass spectral features observed at "low $NO_x$" conditions

$C_5H_{6,8}O_{5-7}$, $C_{6-9}H_{8,10,12,14}O_{6-12}$, $C_{10}H_{14,16,18}O_{5-14}$, and $C_{19-20}H_{28,30,32}O_{9-18}$ ion groups comprised 5%, 36%, 46%, and 4% of the signal detected at "low-$NO_x$" conditions (Fig. 6a), assuming equal CIMS sensitivity and transmission to all detected species . The $C_{10}$ monomers and $C_{19-20}$ dimers compounds that were observed are often associated with atmospheric new particle formation events (Ehn et al., 2014). The prominent $C_{10}H_{14,16}O_{7-9}$ signals detected at m/Q = 308, 310, 324, 326, 340 and 342 in our measurements were dominant signals in previous laboratory and field experiments influenced by the ozonolysis of $\alpha$-pinene emissions (Ehn et al., 2010, 2012, 2014; Jokinen et al., 2015). Other signals that were observed correspond to $C_{5-9}$ species that were generated following carbon-carbon bond cleavage of the $C_{10}$ backbone (Ehn et al., 2012). The remaining $\sim$10% of the signal was classified into $C_{2-3}H_{3,5}NO_5$, $C_5H_7NO_{6-11}$, $C_{6-9}H_{9,11,13,15}NO_{5-10}$, and $C_{10}H_{15,17}NO_{4-14}$ ion groups and may be due to background $NO_x$ in the reactor (Sect. 2.1).

### 3.4.2  $NO_3^-$-CIMS mass spectral features observed at "high $NO_x$" conditions

As was the case with $NO_3^-$-CIMS spectra of isoprene oxidation products, the addition of $N_2O$ to the reactor significantly changed the mass spectrum of $\alpha$-pinene oxidation products (Fig. 6b). At [$N_2O$] = 3.2%, organic nitrates and dinitrates comprised 65% of the total ion signal. We observed reduction in $C_{6-9}H_{8,10,12,14}O_{6-12}$, $C_{10}H_{14,16,18}O_{5-14}$, and $C_{19-20}H_{28,30,32}O_{9-18}$ signals, along with increases in $C_5H_{6,8}O_{5-7}$, $C_5H_7O_{6-11}$, $C_{6-9}H_{9,11,13,15}NO_{5-10}$, $C_{10}H_{15,17}NO_{4-14}$ and $C_{10}H_{16,18}N_2O_{6-13}$ signals. The $C_{10}$ dinitrates may originate from two $\alpha$-pinene + OH reactions followed by two $RO_2$ + NO reactions, but may also include contributions from one $\alpha$-pinene + $NO_3$ reaction followed by one $RO_2$ + NO reaction. The largest signal in Fig. 6b was observed at m/Q = 240, $C_5H_6O_7$. The largest organic nitrate signals in this spectrum were at m/Q = 329, $C_8H_{13}NO_9$, followed by $C_{10}H_{15}NO_9$ (m/Q = 355), $C_{10}H_{16}N_2O_9$ (m/Q = 354), and $C_{10}H_{15}NO_8$ (m/Q = 339).

Figure 6c shows $C_5H_6O_{5-7}$, $C_{6-9}H_{8,10,12,14}O_{6-12}$, $C_{10}H_{14,16,18}O_{5-14}$, $C_{19-20}H_{28,30,32}O_{9-18}$, $C_{2-3}H_{3,5}NO_5$, $C_5H_7NO_{6-11}$, $C_{6-9}H_{9,11,13,15}NO_{5-10}$, $C_{10}H_{15,17}NO_{4-14}$, and $C_{10}H_{16,18}N_2O_{6-13}$ signals detected with $NO_3^-$-CIMS at the Centreville site during the SOAS campaign. The spectra shown here were obtained during the sampling period shown in Fig. 4f and, given the large number of compounds, may include contributions from HOM precursors other than $\alpha$-pinene. A comparison of Figs. 6a-6c indicates that adding $N_2O$ to the reactor increases the similarity between the composition of $\alpha$-pinene oxidation products generated in the reactor and under "high-NO" ambient conditions, especially in regards to the enhanced $C_5H_6O_{5-7}$, $C_{6-9}H_{9,11,13,15}NO_{5-10}$, $C_{10}H_{15,17}NO_{4-14}$, and $C_{10}H_{16,18}N_2O_{6-13}$ signals.

### 3.4.3  Detection of acylperoxy nitrates from $RO_2$ + $NO_2$ reactions

Figs. 6b and 6c indicate that PAN (m/Q = 183, $C_2H_3NO_5$) and PPN (m/Q = 197, $C_3H_5NO_5$) are formed at lower yields (<0.4%) than were observed with isoprene (Fig. 4c and 4d), suggesting that PAN and PPN formation from reaction of $\alpha$-pinene-

derived–$RO_2$ with $NO_2$ are not enhanced in the reactor compared to atmospheric conditions. $C_9H_{13}NO_6$ and $C_{10}H_{15}NO_{6-8}$ are APNs generated following OH oxidation of pinonaldehyde, a major first-generation oxidation product of $\alpha$-pinene, with
termination by $RO_2$ + $NO_2$ reaction (Eddingsaas et al., 2012). All four compounds are detected in the reactor and ambient $NO_3^-$-CIMS spectra shown in Figs. 6b and 6c, with $C_{10}H_{15}NO_{6-8}$ signals among the largest in the spectra. If these signals represent APNs, they appear to be important in both laboratory and atmospheric conditions.

### 3.4.4  Influence of $\alpha$-pinene + $NO_3$ reactions

Our calculations suggest that $\alpha$-pinene + $NO_3$ reactions compete with $\alpha$-pinene + OH reactions at the experimental conditions
used to generate the $NO_3^-$–CIMS spectrum shown in Fig. 6b (Fig. S7). If this were the case, enhanced yields of $C_{10}H_{15}NO_6$ are anticipated from $\alpha$-pinene + $NO_3$ reaction to generate pinonaldehyde, followed by pinonaldehyde + $NO_3$ reaction and $RO_2$ + $NO_2$ termination (Perraud et al., 2010; Nah et al., 2016) Other minor $\alpha$-pinene + $NO_3$ products detected with CIMS include $C_{10}H_{15}NO_5$, $C_9H_{13}NO_6$, $C_{10}H_{16}N_2O_7$, and $C_{10}H_{15}NO_9$ (Nah et al., 2016). We hypothesize that if $\alpha$-pinene + $NO_3$ reactions influence the spectrum shown in Fig. 6b, $C_{10}H_{15}NO_6 : C_{10}H_{15}NO_8$ should be higher in Fig. 6b than in Fig. 6c.
Instead, the $C_{10}H_{15}NO_6 : C_{10}H_{15}NO_8$ ratio was 0.12 in the reactor and 0.28 at the Centreville site during a daytime period (0730 - 1100) with presumably negligible $NO_3$ influence.

   Dinitrates ($C_{10}H_{16,18}N_2O_{6-13}$) shown in Fig. 6b may originate from two $\alpha$-pinene + OH reactions followed by two $RO_2$ + NO terminations, or one $\alpha$-pinene + $NO_3$ reaction followed by one $RO_2$ + NO termination. Given comparable calculated OH and $NO_3$ reaction rates under these conditions (Fig. S7e), we hypothesize that the majority of dinitrate signals should
originate from $\alpha$-pinene + $NO_3$ reactions if their yields are not oxidant-dependent. If this were the case, $C_{10}H_{16,18}N_2O_{6-13}$: $C_{10}H_{15,17}NO_{4-14}$ should be larger in Fig. 6b than in Fig. 6c. However, $C_{10}H_{16,18}N_2O_{6-13}$: $C_{10}H_{15,17}NO_{4-14}$ was 0.23 in the reactor and 0.61 at the Centreville site. Thus, while the calculated $\alpha$-pinene + $NO_3$ oxidation rate is significant (Fig. S7e), it is not clear that $\alpha$-pinene + $NO_3$ oxidation products significantly affect the spectrum shown in Fig. 6b. This may be due to significantly lower organic nitrate yields from $\alpha$-pinene + $NO_3$ than from $\alpha$-pinene + OH reactions in the presence of NO (Fry
et al., 2014; Rindelaub et al., 2015).

### 3.5  Transition from $RO_2$+$HO_2$ to $RO_2$+NO-dominant regimes observed in isoprene and $\alpha$-pinene oxidation products

Figures 7 and 8 shows normalized signals of the representative groups of isoprene and $\alpha$-pinene oxidation products as a function of increasing NO:$HO_2$, which may be influenced by NO + $HO_2$, NO + $RO_2$ and $HO_2$ + $RO_2$ reactions in the reactor. For each
group of compounds, signals obtained at a specific NO:$HO_2$ were normalized to the maximum observed signal. NO:$HO_2$ is correlated with the relative branching ratios of $RO_2$ + $HO_2$ and $RO_2$ + NO reactions that govern the distribution of oxidation products observed in Figs. 4 and 6. As is evident from Figs. 7 and 8, different ion families were characterized by different trends as a function of NO:$HO_2$. The normalized signals of $C_{4-5}$ (isoprene), $C_{6-10}$ ($\alpha$-pinene) and $C_{19-20}$ ($\alpha$-pinene) species decreased monotonically with increasing NO:$HO_2$. In Fig. 8, the abundance of $C_{19-20}$ dimers decreased significantly faster

than the $C_{6-10}$ species. Because dimers are products of $RO_2$ + $RO_2$ self-reactions, their yield is quadratic with respect to $[RO_2]$ and therefore was more affected by competing $RO_2$ + NO reactions than species formed from $RO_2$ + $HO_2$ reactions.

The normalized signals of $C_5$ (isoprene) and $C_{10}$ ($\alpha$-pinene) organic nitrates reached their maximum values at $NO:HO_2 \approx$ 1 prior to decreasing. Maximum signals of $C_{6-9}$ organic nitrates ($\alpha$-pinene) were obtained at $NO:HO_2$ = 2.4, and maximum signals of $C_5$ (isoprene) and $C_{10}$ ($\alpha$-pinene) dinitrates were obtained at $NO:HO_2$ = 5.2 and 6.4. The formation of dinitrates was favored when $RO_2$ + NO >> $RO_2$ + $HO_2$, as expected, and regardless of whether $RO_2$ was formed from oxidation of $\alpha$-pinene by OH, $O_3$ or $NO_3$. We hypothesize that $NO:HO_2$ >> 1 favored $RO_2$ + NO $\rightarrow$ RO + $NO_2$ fragmentation reactions that led to formation of smaller, more volatile $C_5H_{6-8}O_{5-7}$ and $C_5H_7NO_{6-11}$ $\alpha$-pinene oxidation products (Atkinson, 2007; Chacon-Madrid and Donahue, 2011), whose signals continuously increased with increasing $NO:HO_2$, along with other products not detected with $NO_3^-$-CIMS. This pathway apparently competed with $RO_2$ + NO $\rightarrow$ $RO_2NO$ reactions that led to formation of $C_5$ isoprene dinitrates, $C_6$-$C_{10}$ $\alpha$-pinene nitrates and $C_{10}$ $\alpha$-pinene dintrates.

Isoprene oxidation products such as $C_5H_9NO_7$ and $C_5H_{11}NO_7$ contain one peroxide and one nitrate functional group, and $C_5H_9NO_8$ contains two peroxides and one nitrate functional group. The formation of these species, as well as $C_{6-10}$ $\alpha$-pinene-derived organic nitrates, was favored at $NO:HO_2 \approx$ 1-2 where the relative rates of $RO_2$ + NO and $RO_2$ + $HO_2$ reactions were similar. This correlation suggests that the $C_{6-10}$ $\alpha$-pinene organic nitrates detected with $NO_3^-$-CIMS contained a combination of peroxide and nitrate functional groups, whereas $C_5$ (isoprene) and $C_{10}$ ($\alpha$-pinene) dinitrates contained fewer functional groups that were specifically formed from $RO_2$ + $HO_2$ reactions.

## 4    Atmospheric Implications

The use of $O(^1D)$ + $N_2O$ reactions facilitates systematic control of $NO:HO_2$ over the range of "$RO_2$ + $HO_2$ dominant" to "$RO_2$ + NO dominant" conditions. Further, this is accomplished with the use of a single OH radical precursor ($O_3$) that has previously hindered characterization of $NO_x$-dependent chemistry in oxidation flow reactors. Our results suggest that this method can be used to identify molecular tracers for processes influenced by $RO_2$ + NO and/or $RO_2$ + $NO_2$ reactions (Figs. 4 and 6). This method will be used in future work to investigate the influence of $NO_x$ on physicochemical properties of secondary organic aerosols such as hygroscopicity and refractive indices over an atmospherically relevant range of $NO:HO_2$. Care should be taken to use experimental conditions that minimize the relative contributions of unwanted $NO_3$-initiated oxidation chemistry – for example, $[O_3]$ >> 5 ppm and $[H_2O]$ << 1% (Figs. S2 and S3) – particularly when using species such as $\alpha$-pinene that are highly reactive to $NO_3$. While potential formation of dinitrates from $\alpha$-pinene + $NO_3$ reactions at high-NO conditions was not the primary goal of this experiment, we note that this chemical fingerprint has been observed in ambient measurements (Yan et al., 2016) and thus represents an additional application of $O(^1D)$ + $N_2O$ reactions in future work. Additionally, studies that require multiple days of equivalent atmospheric OH oxidation at $NO:HO_2$ >>1 should consider implementing 185 nm photolysis of $H_2O$ and $N_2O$ to provide additional sources of $O(^1D)$ and OH that may decrease OH suppression due to competing $O(^1D)$ + $H_2O$ and $O(^1D)$ + $N_2O$ reactions.

*Acknowledgements.* This research was supported by the Atmospheric Chemistry Program of the U.S. National Science Foundation under grants AGS-1536939, AGS-1537446 and AGS-1537009 and by the U.S. Office of Science (BER), Department of Energy (Atmospheric Systems Research) under grants DE-SC0006980 and DE-SC0011935. A. T. Lambe thanks Gabriel Isaacman-VanWertz and Jesse Kroll (Massachusetts Institute of Technology) for helpful discussion, Zhe Peng and Jose Jimenez (University of Colorado - Boulder) for preliminary input on method operation with added $\lambda = 185$ nm radiation, and Karsten Baumann and Eric Edgerton (Atmospheric Research and Analysis, Inc.) for the use of ambient NO measurements during the SOAS campaign.

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

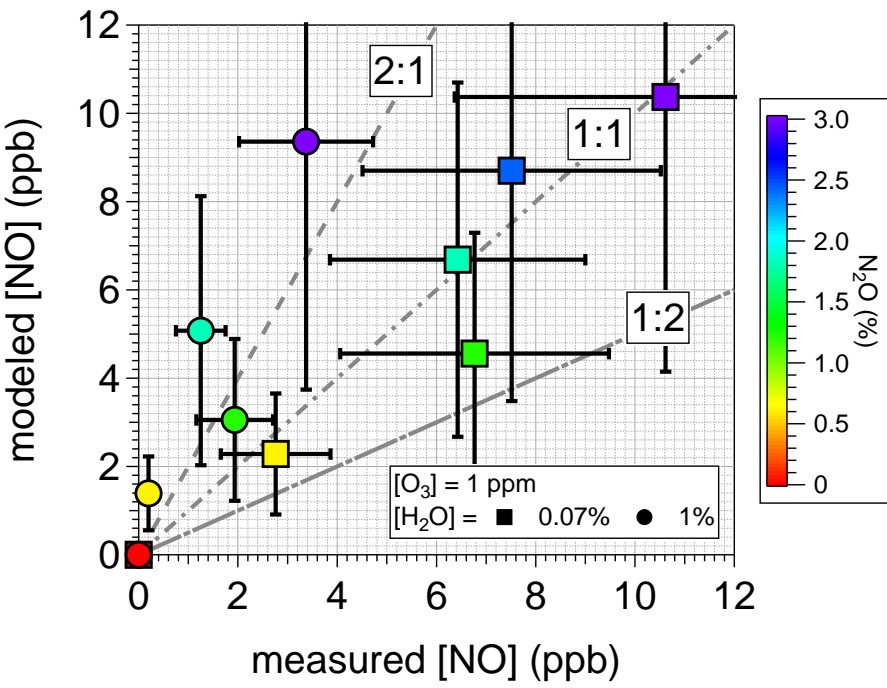

**Figure 1.** Scatter plot comparing measured and modeled [NO] at 80 sec residence time in the PAM oxidation flow reactor, $I_{254} = 4 \times 10^{15}$ ph cm$^{-2}$ sec$^{-1}$, $[O_3]$ = 1 ppm, $[H_2O]$ = 0.07 and 1%, $[N_2O]$ = 0 to 3%. Symbols are colored by $[N_2O]$, with 1:2, 1:1 and 2:1 lines shown for reference. Error bars represent ±60% uncertainty in model outputs (Peng et al., 2015) and ±40% precision in replicate [NO] measurements at fixed $[N_2O]$.

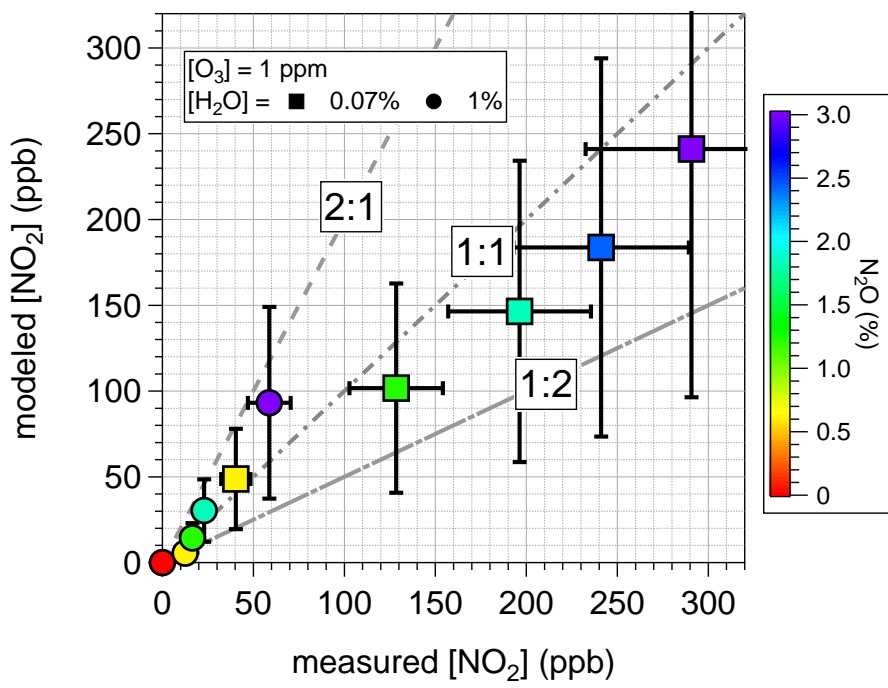

**Figure 2.** Scatter plot comparing measured and modeled [$NO_2$] at 80 sec residence time in the PAM reactor, $I_{254}$ = 4×$10^{15}$ ph cm$^{-2}$ sec$^{-1}$, [$O_3$] = 1 ppm, [$H_2O$] = 0.07 and 1%, [$N_2O$] = 0 to 3%. Symbols are colored by [$N_2O$], with 1:2, 1:1 and 2:1 lines shown for reference. Error bars represent ±60% uncertainty in model outputs (Peng et al., 2015) and ±20% precision in replicate [NO] measurements at fixed [$N_2O$].

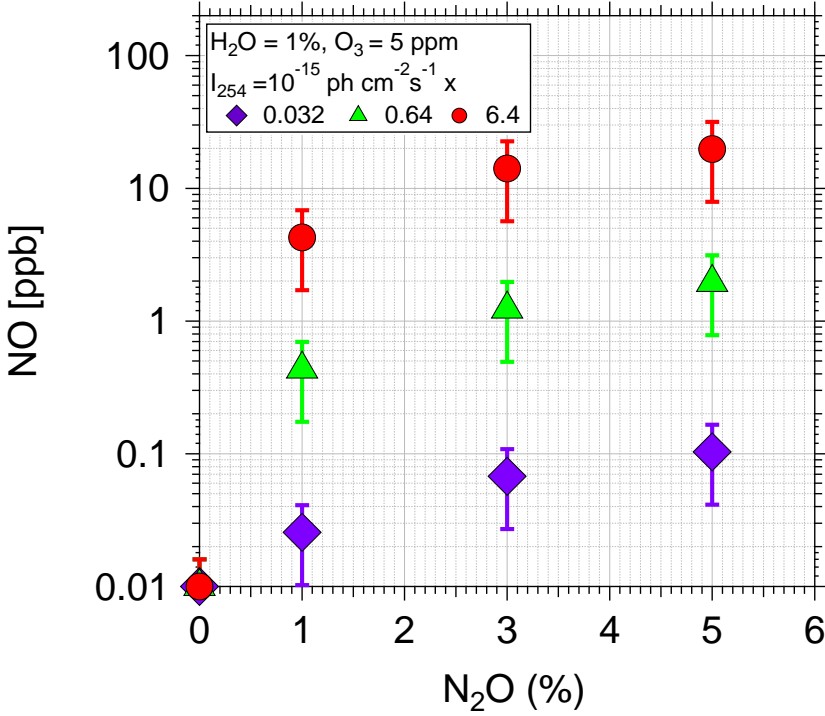

**Figure 3.** Modeled steady-state [NO] as a function of [$N_2O$] input to the PAM reactor at $I_{254}$ = 0.032×10$^{15}$, 0.64×10$^{15}$ and 6.4×10$^{15}$ ph cm$^{-2}$ sec, [$H_2O$] = 1%, [$O_3$] = 5 ppm, mean residence time = 80 sec. Error bars represent ± 60% uncertainty in modeled [NO] (Peng et al., 2015).

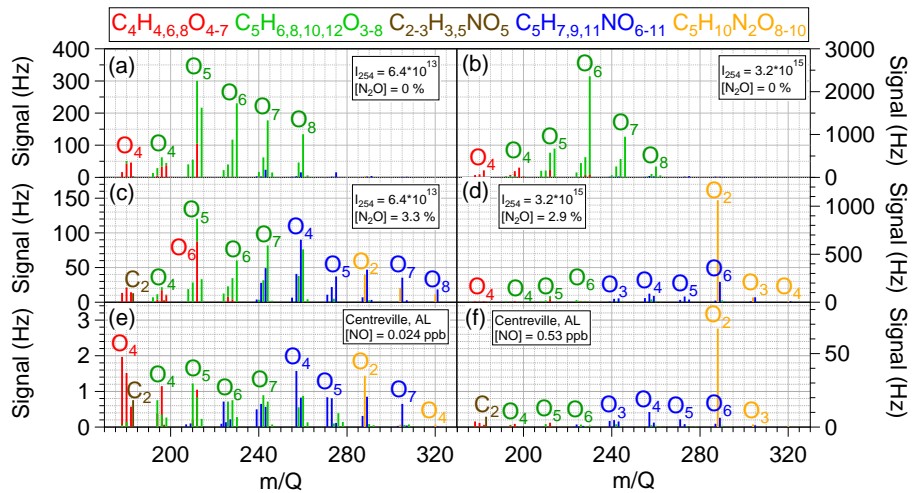

**Figure 4.** $NO_3^-$-CIMS mass spectra of isoprene oxidation products generated in the PAM reactor at $[H_2O]$ = 1%, $[O_3]$ = 5 ppm, mean residence time = 80 sec: (a) $I_{254}$ = 6.4×$10^{13}$ ph cm$^{-2}$ sec$^{-1}$, $[N_2O]$ = 0%; (b) $I_{254}$ = 3.2×$10^{15}$ ph cm$^{-2}$ sec$^{-1}$, $[N_2O]$ = 0 %; (c) $I_{254}$ = 6.4×$10^{13}$ ph cm$^{-2}$ sec$^{-1}$, $[N_2O]$ = 3.2 %; (d) $I_{254}$ = 3.2×$10^{15}$ ph cm$^{-2}$ sec$^{-1}$, $[N_2O]$ = 2.9 %. $NO_3^-$-CIMS mass spectra of the same compounds detected at the SOAS ground site in Centreville, Alabama, USA during (e) "low-NO" and (f) "high-NO" conditions (see text for additional details; $C_5H_6O_{5-7}$ ions removed from SOAS spectra due to larger contributions from $\alpha$-pinene + OH oxidation products (Fig. 6). "$C_x$" or "$O_x$" indicates number of carbon or atoms in labeled ions (not including oxygen atoms in nitrate functional groups).

**Figure 5.** Simplified reaction scheme summarizing known isoprene + OH reaction pathways yielding multigeneration oxidation products. Four peroxy radical ($RO_2$) isomers are generated following initial OH addition to isoprene: $1,2\text{-}RO_2$, $4,3\text{-}RO_2$, $1,4\text{-}RO_2$, $4,1\text{-}RO_2$. The $1,2\text{-}RO_2$ and $4,3\text{-}RO_2$ isomers follow the same reaction pathways, yielding chemical formulas with green text that were detected with $NO_3^-$-CIMS. The $4,1\text{-}RO_2$ isomer yields $C_5H_{10}O_7$, also detected with $NO_3^-$-CIMS. Chemical formulas with red text may be generated through the methacrolein (MACR) channel but were not detected with $NO_3^-$-CIMS .

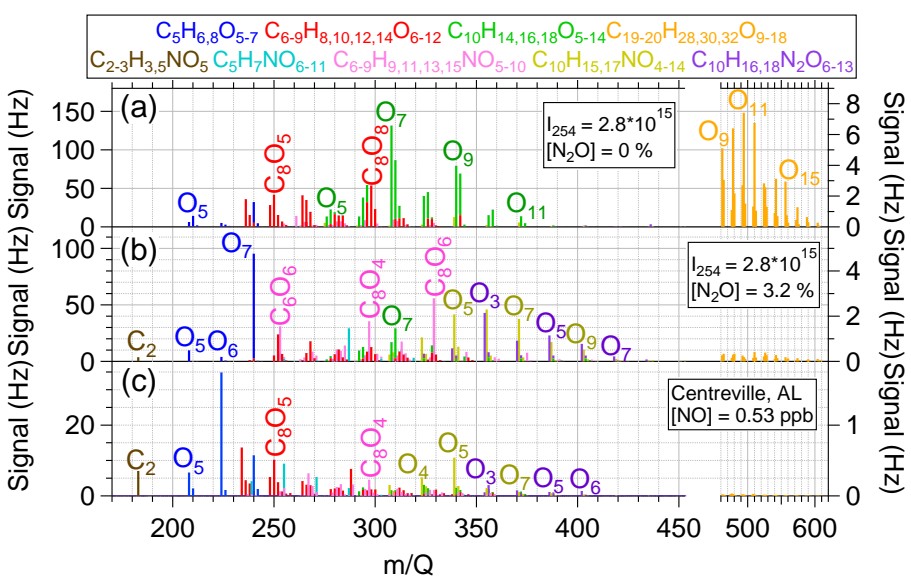

**Figure 6.** $NO_3^-$-CIMS mass spectra of $\alpha$-pinene oxidation products generated in the PAM reactor at $[H_2O] = 0.07\%$, $[O_3] = 5$ ppm, mean residence time = 80 sec: (a) $I_{254} = 2.8 \times 10^{15}$ ph cm$^{-2}$sec$^{-1}$, $[N_2O] = 0$ %; (b) $I_{254} = 2.8 \times 10^{15}$ ph cm$^{-2}$sec$^{-1}$, $[N_2O] = 3.2$ %. (c) $NO_3^-$-CIMS mass spectra of the same compounds detected at the SOAS ground site in Centreville, Alabama, USA during "high-NO" conditions shown in Fig. 4f (note: $C_5H_7NO_{6-11}$ signals in SOAS spectra also contributed from isoprene + OH oxidation products). "$C_x$" or "$O_x$" labels indicate number of oxygen atoms in corresponding signals (not including oxygen atoms in nitrate functional groups).

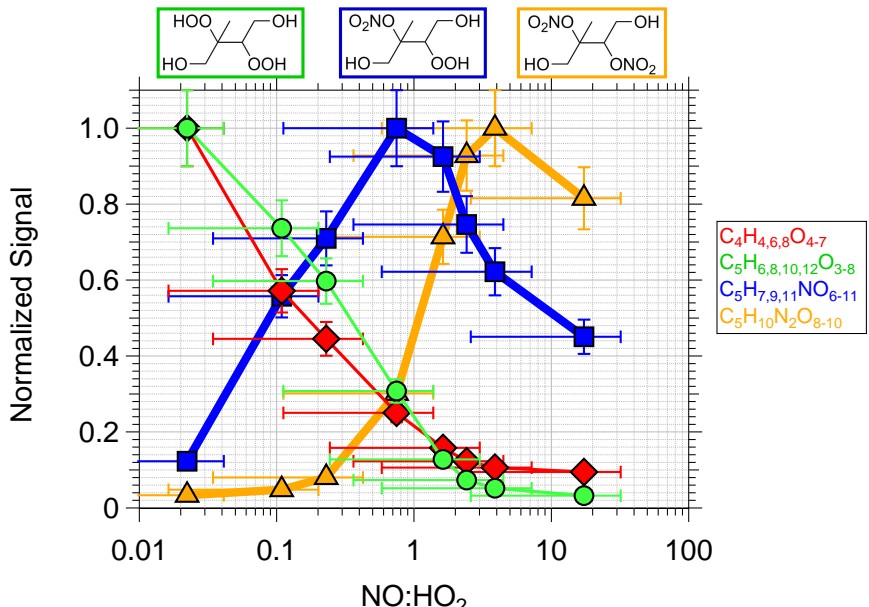

**Figure 7.** Normalized $NO_3^-$-CIMS signals of $C_4H_{4,6,8}O_{4-7}$, $C_5H_{6,8,10,12}O_{3-8}$, $C_5H_{7,9,11}NO_{6-11}$, and $C_5H_{10}N_2O_{8-10}$ isoprene oxidation products generated in the PAM reactor at $I_{254} = 3.2 \times 10^{15}$ ph cm$^{-2}$ sec$^{-1}$, [$H_2O$] = 1%, [$O_3$] = 5 ppm, mean residence time = 80 sec as a function of modeled $NO:HO_2$. For each of the species classes, signals were normalized to the maximum signal. Proposed structures for $C_5H_{12}O_6$, $C_5H_{11}NO_7$, and $C_5H_{10}N_2O_8$ signals are shown as representative compounds for each species class (Fig. 5). Representative error bars indicate $\pm 1\sigma$ uncertainty in $NO_3^-$-CIMS signals and $\pm 85\%$ uncertainty in $NO:HO_2$.

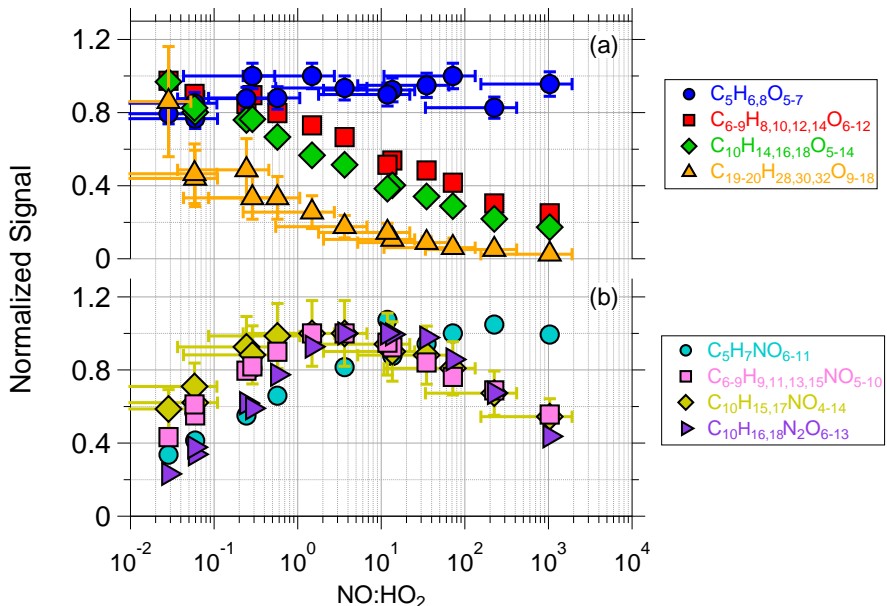

**Figure 8.** Normalized $NO_3^-$-CIMS signals of (a) $C_5H_{6,8}O_{5-7}$, $C_{6-9}H_{8,10,12,14}O_{6-12}$, $C_{10}H_{14,16,18}O_{5-14}$, $C_{19-20}H_{28,30,32}O_{9-18}$ and (b) $C_5H_7NO_{6-11}$, $C_{6-9}H_{9,11,13,15}NO_{5-10}$, $C_{10}H_{15,17}NO_{4-14}$, and $C_{10}H_{16,18}N_2O_{6-13}$ $\alpha$-pinene oxidation products generated in the PAM reactor at $I_{254}$ = 2.8×10$^{15}$ ph cm$^{-2}$ sec$^{-1}$, [$H_2O$] = 0.07%, [$O_3$] = 5 ppm, mean residence time = 80 sec as a function of modeled NO:HO$_2$. For each of the species classes, signals were normalized to the maximum signal. Representative error bars indicate $\pm$ 1$\sigma$ uncertainty in $NO_3^-$-CIMS signals and $\pm$ 85% uncertainty in modeled NO:HO$_2$.