# Peer review of "Controlled nitric oxide production via $O(^1D)+N_2O$ reactions for use in oxidation flow reactor studies"

_Atmospheric Measurement Techniques, 2016_

## Referee Comment (RC1) · Anonymous Referee #1 · 6 Feb 2017

The manuscript, "Controlled nitric oxide production via O(1D)+N2O reactions for use in oxidation flow reactor studies" by Lambe et al. presents a creative concept for generating and maintaining a more realistic level of NO in atmospheric simulation experiments. The work is presented neatly, figures constructed meticulously. The idea presented here would be useful for the atmospheric chemistry community, but in the way the technique is currently described, the advantages gained from this technique over previous ones do not seem like significant progress. The work deserves publication following some revisions and clarifications.

The main issue I find with this work is that it completely missed the opportunity to demonstrate the effectiveness of this newly proposed technique with actual measurements of NO and N2O. This seems particularly egregious given the moderate range of NO modeled here (figure 1) that is not difficult to measure directly. Was a NO instrument not available when this work was conceived? Wasn't producing and maintaining a predictable amount of NO the objective here? Why not show that you achieved this with observations, in addition to model? Moreover, measurement of N2O, specifically, the decrease in N2O mixing ratio with increasing radiation intensity would have been helpful to ensure that N2O reaction and photolysis were indeed the reason for the observed changes in CIMS spectra. These two measurements of NO and N2O would have been more convincing than the results from the CIMS, which I find less than convincing, if not, altogether unnecessary.

The level of O3 (500 ppb to 50 ppm) required to generate enough O(1D) is still much too high to simulate anything that resembles atmospherically realistic conditions. Limitations are two-fold that I can think of: (1) Given ppm levels of O3, oxidation by ozonolysis can compete with OH oxidation making systematic study of one oxidation pathway versus the other difficult. (2) NO to NO2 ratio also deviates from ambient, such that oxidation by NO3 radical becomes non-negligible (along with production of HNO3, peroxyacetyl nitrates, etc.). This, the authors note could be as high as 40% of the total oxidation by OH, O3 and NO3 combined. Perhaps the BVOC products from OH vs O3 vs NO3 can be separated using CIMS data, but the presence of different RO2 isomers resulting from each oxidant that may react with one another (RO2+RO2) may mean this simple attribution may not be possible, particularly if the products possess different functional group but same molecular composition. The ppm O3 levels used here also seem at odds with statements in the abstract and elsewhere in the manuscript that seem to suggest that ppm levels of O3 are bad (lines 1-5), and that this proposed technique doesn't require ppm O3 levels. In any case, have the authors attempted to run the chamber without O3? What is the highest level of NO achieved by percent levels of N2O due to direct photolysis? Given that most suburban+rural+remote regions experience highest NO levels less than 0.5 ppb, perhaps just N2O and ambient level of O3 would suffice? This approach may not be suitable for flow reactors, but it may be

for the more traditional atmospheric simulation chamber studies.

The CIMS data show that, yes, as you increase NO, the level of organic nitrates increase, and levels of most organics without a nitrate decreases. This is not surprising. What is missing is that this method of NO generation by N2O in the flow reactor can demonstrate atmospherically relevant chemistry. If CIMS is the instrument of choice, the authors need to compare CIMS spectra of the flow reactor and one that was obtained from ambient atmosphere.

The discussion sections on the types of oxidation products observed are less than convincing, lacking the detailed mechanism discussions typically included in such studies. As such, these sections read more like speculation. Does the model account for RO2 chemistry? Is there a model output for the various organic molecular compositions observed or at least groups of organics (i.e. Krechmer 2015 ES&T)?

How much of NO:HO2 changes (x-axis; figures 4 and 5) are due to the reaction of NO with HO2? Is RO2 accounted for in the calculation of NO and HO2?

Figure 2, judging from the y-axes, much higher signal levels are observed at higher I254. Is this the result of production of later-generation oxidation products? Or just more complete oxidation? Was the amount of parent BVOC oxidized measured?

Figure 4 is misleading. From what I understand, the CIMS identifies molecular compositions but cannot assign structure/isomer/functional groups. What is the source of the drawings on top of figure 4? How were they determined?
* * *

---

## Author Comment (AC1)

We thank the reviewers for their valuable input on the manuscript, and we thank Dr. Hanisco for serving as editor and overseeing the review process. The response to each reviewer comment (red text) is provided below (blue text). Following each response, revisions to manuscript text are indicated in highlighted black text.

**Reviewer #1**

1. The main issue I find with this work is that it completely missed the opportunity to demonstrate the effectiveness of this newly proposed technique with actual measurements of NO and N2O. This seems particularly egregious given the moderate range of NO modeled here (figure 1) that is not difficult to measure directly. Was a NO instrument not available when this work was conceived? Wasn't producing and maintaining a predictable amount of NO the objective here? Why not show that you achieved this with observations, in addition to model?

**Response**. In an early stage of method development, we conducted a set of experiments where NO and $NO_2$ were measured following $N_2O$ (0-3%), $O_3$ (1 ppm) and $H_2O$ (0.07 – 1%) addition to the reactor. The measurements and their interpretation are not straightforward and are of limited use above $[O_3]$ ~ 1 ppm, for reasons that are discussed below. However we agree that conveying this information in the manuscript would be useful, and would help to motivate the use of the photochemical model to explore a wider range of operating conditions.

We will add a new Section 3.1 describing the measured NO and $NO_2$ mixing ratios and comparison with photochemical model outputs. Two accompanying figures, Figure 1 and Figure 2 (below), will be added to the revised manuscript. Subsequent subsections and figures will be renumbered accordingly.

**Revision to Section 2.1.** Changes below incorporate earlier test conditions.

"$O_3$ (~1-5 ppm) was generated outside the flow reactor by $O_2$ irradiation at 185 nm using a mercury lamp. $O(^1D)$ was produced by photolysis of $O_3$ at 254 nm inside the reactor using two or four mercury lamps"

**Revision to Section 2.2**.

"2.2 $NO_x$ and chemical ionization mass spectrometer (CIMS) measurements

In one set of experiments, [NO] and [$NO_2$] were measured downstream of the reactor with a Thermo Scientific Model 42i chemiluminescent analyzer and an Aerodyne Cavity Attenuated Phase Shift (CAPS) $NO_2$ analyzer, which measures $NO_2$ absorption at $\lambda$ = 450 nm (Kebabian et al., 2008). During these experiments, the following operating conditions were used: $I_{254}$ = 4*10$^{15}$ ph cm$^{-2}$ sec$^{-1}$, $[O_3]$ = 1

ppm, $[H_2O]$ = 0.07% and 1%, $[N_2O]$ = 0 to 3%. These conditions assess a subset of the attainable operating conditions for comparison with outputs of the photochemical model described in Section 2.3.

The measured $NO_2$ mixing ratio was decreased by 10 ppb due to absorption by 1 ppm $O_3$ at 450 nm in the absence of $NO_2$. The measured NO mixing ratio was scaled by a factor of 3.2 for depletion downstream of the reactor due to 1.2 sec reaction time with 1 ppm $O_3$ in the sample line, assuming $k_{NO+O3}$ = $1.8 \times 10^{-14}$ $cm^3$ $molec^{-1}$ $sec^{-1}$ and pseudo-first order conditions (Atkinson et al., 2004). Additional NO depletion inside the Thermo analyzer (~47% at 1 ppm $O_3$) was accounted for in a separate experiment where known mixing ratios of NO (50 ppb) and $O_3$ (0 to 6.9 ppm) were added at the inlet of the instrument (Fig. S1). Because the combined NO depletion in the sample line and the NO analyzer is significantly higher at higher $[O_3]$ (e.g. ~90% at $[O_3]$ = 2 ppm and ~99.6% at $[O_3]$ = 5 ppm), accurate experimental characterization of [NO] is more difficult above $[O_3]$ ~ 1 ppm.

In another set of experiments, mass spectra of isoprene and α-pinene gas-phase oxidation products were obtained with an Aerodyne high-resolution time-of flight mass spectrometer (Bertram et al.,2011)…."

**Revisions to Section 3.**

**3.1. Comparison of measured and modeled [NO] and [NO₂] values following O(¹D) + N₂O and NO + O₃ reactions**

Figure 1 compares modeled and measured NO mixing ratios obtained following 80 sec residence time in the reactor at the operating conditions described in Sect. 2.2. The corresponding integrated OH exposures are approximately $2.6 \times 10^{11}$ and $2.4 \times 10^{12}$ molec $cm^{-3}$ sec, respectively, in the absence of added $N_2O$. Symbols are colored by $[N_2O]$ which ranged from 0 to 3%.Measured [NO] ranged from 0 to 10.4 ppb and increased with increasing $[N_2O]$, as expected, at both $[H_2O]$ = 0.07% and 1%. The mean ratio of modelled-measured [NO] was 0.94 ± 0.19 at $[H_2O]$ = 0.07% and 3.85 ± 2.33 at $[H_2O]$ = 1%.

[Figure]

**Figure 1**. Scatter plot comparing measured and modeled [NO] at 80 sec residence time in the PAM oxidation flow reactor, $I_{254}$ = 4*10$^{15}$ ph cm$^{-2}$ sec$^{-1}$, [O$_3$] = 1 ppm, [H$_2$O] = 0.07 and 1%, [N$_2$O] = 0 to 3%. Symbols are colored by [N$_2$O], with 1:2, 1:1 and 2:1 lines shown for reference. Error bars represent ±60% uncertainty in model outputs (Peng et al., 2015) and ±40% precision in replicate [NO] measurements at fixed [N$_2$O].

[Figure]

**Figure 2**. Scatter plot comparing measured and modeled [NO$_2$] at 80 sec residence time in the PAM oxidation flow reactor, $I_{254}$ = 4*10$^{15}$ ph cm$^{-2}$ sec$^{-1}$, [O$_3$] = 1 ppm, [H$_2$O] = 0.07 and 1%, [N$_2$O] = 0 to 3%. Error bars represent ±60% uncertainty in model outputs (Peng et al., 2015) and ±20% precision in replicate [NO$_2$] measurements at fixed [N$_2$O].

NO$_2$, which is formed by the NO + O$_3$ reaction, is more straightforward to measure under these conditions because NO$_2$ reacts approximately 500 times slower than NO with O$_3$. Thus, a comparison of modeled and measured [NO$_2$] values provides additional method evaluation with less uncertainty than [NO] measurements. Figure 2 compares corresponding modeled and measured NO$_2$ mixing ratios obtained during the same experiments described in Figure 1. As expected, [NO$_2$] increased with increasing [N$_2$O] because of faster NO + O$_3$ reaction rate from increasing [NO]. At [H$_2$O] = 0.07%, measured [NO$_2$] ranged from 0 to 291 ppb, whereas at [H$_2$O] = 1%, measured [NO$_2$] ranged from 0 to 59 ppb. [NO$_2$] was lower in the latter case because additional OH was formed from O($^1$D) + H$_2$O reactions (Section 2.1), which increased the rate of the OH + NO$_2$ reaction. The mean ratio of modelled-measured [NO$_2$] was 0.72 ± 0.39 at [H$_2$O] = 0.07% and 1.05 ± 0.50 at [H$_2$O] = 1%. These results, combined with results shown in Figure 1, suggest that an uncharacterized H$_2$O- or HNO$_3$-related artifact negatively biased the measured [NO] values at [H$_2$O] = 1%, and that the photochemical model described in Section 2.3 may be used to evaluate a wider range of reactor operating conditions.

**Revision to Supplement.** We will add a new Figure S1 (below) to the revised manuscript supplement. Subsequent figures in the Supplement will be renumbered accordingly.

[Figure]

**Figure S1**. NO depletion inside the NO analyzer due to reaction of 50 ppb initial NO (NO$_i$) with O$_3$. NO was introduced from a calibration cylinder, and O$_3$ was introduced from the output of the PAM reactor.

**Revision to References**. The following citations will be added.

Atkinson, R.; Baulch, D.L.; Cox, R.A.; Crowley, J.N.; Hampson, R.F.; Hynes, R.G.; Jenkin, M.E.; Rossi, M.J.; Troe, J. Evaluated kinetic and photochemical data for atmospheric chemistry: Volume I - gas phase reactions of Ox, HOx, NOx and SOx species. Atmos. Chem. Phys., 4,  1461 – 1738, 2004 .

P.L. Kebabian, E.C. Wood, S.C. Herndon, A. Freedman. A Practical Alternative to Chemiluminescence Detection of Nitrogen Dioxide: Cavity Attenuated Phase Shift Spectroscopy**,** Environ. Sci. Technol., 42, 6040-6045, 2008.

2. Moreover, measurement of N2O, specifically, the decrease in N2O mixing ratio with increasing radiation intensity would have been helpful to ensure that N2O reaction and photolysis were indeed the reason for the observed changes in CIMS spectra. These two measurements of NO and N2O would have been more convincing than the results from the CIMS, which I find less than convincing, if not, altogether unnecessary.

**Response.** This suggestion is very difficult to implement, with minimal added benefit compared to the CIMS spectra and the [NO] measurements described above. First, please note $N_2O$ photolysis is negligible at 254 nm: the main loss pathway under the conditions used in these measurements is $N_2O + O(^1D)$. Second, measuring the decrease in $[N_2O]$ due to reaction with $O(^1D)$ requires a nearly unattainable level of precision. For example, at 80 sec residence time in the reactor, assuming inputs of $[N_2O]$ = 1000 ppb, $[H_2O]$ = 1%, and $[O_3]$ = 5 ppm, the modeled $[N_2O]$ at the exit of the reactor is $[N_2O]$ = 999.96 ppb, corresponding to $\Delta[N_2O]$ = -0.04 ppb or a necessary precision of 0.04/1000 = 0.004%.

**Revision**. We do not think manuscript revisions are necessary in response to this comment.

3. The level of O3 (500 ppb to 50 ppm) required to generate enough O(1D) is still much too high to simulate anything that resembles atmospherically realistic conditions. Limitations are two-fold that I can think of: (1) Given ppm levels of O3, oxidation by ozonolysis can compete with OH oxidation making systematic study of one oxidation pathway versus the other difficult.

**Response.** This is a common misconception. It is true that for some species, and/or non-judicious operating conditions, ozonolysis is too fast to allow the systematic study of OH oxidation difficult. Otherwise, in many cases oxidation by OH remains the dominant loss pathway. We refer the reviewer to the detailed

discussion of this issue in Peng et al. (2016), including the following discussion from Section 3.1.5 of Peng et al. (2016) that is quoted below:

"Among the literature OFR studies, the field studies employing OFRs in urban and forested areas all operated under $O_{3exp}/OH_{exp}$ values 100 times lower than in the atmosphere. In these field studies reaction of almost all VOCs with $O_3$ can be neglected, except for the most reactive biogenics with $O_3$, e.g., α-terpinene and β-caryophyllene. The source study in an urban tunnel of Tkacik et al. (2014) operated under similar conditions. Some laboratory studies using OFR254 (Kang et al., 2011; Lambe et al., 2011b) as well as the biomass smoke source study (Ortega et al., 2013) operated at $O_{3exp}/OH_{exp}$ close to tropospheric values, because the injected $O_3$ plays a key role for OFR254 studies and the biomass smoke experiments were conducted at high $OHR_{ext}$."

We will modify Figures S4d, S5d and S6e in the Supplement to indicate the relative roles of OH, $O_3$ and $NO_3$ for oxidation of isoprene or $\alpha$-pinene under the experimental conditions that are used. In these figures, the fraction loss of VOC to each oxidant is determined from the integrated OH, $O_3$ and $NO_3$ exposure (calculated using the photochemical model described in Sect. 2.3) and corresponding published OH, $O_3$ and $NO_3$ rate constants. This axis replaces the "OH:NO3" axis shown previously in Figs. S4d, S5d and S6e. As an example, the revised Figure S4 is shown below.

**Revision to Section 2.1**. "In most cases, oxidation of VOCs by $O_3$ is slower than oxidation by OH radical, even with parts per million levels of $O_3$ present (e.g. Peng et al., 2016). $NO_3$ radicals, which are produced as a byproduct of $NO_2 + O_3$ or $HNO_3 + OH$ reactions, can potentially convolute interpretation of results if the relative oxidation rates of isoprene/α-pinene by OH and $NO_3$ are comparable...."

**Revision to Figure S4** (revised Figures S5-S6 similar format):

[Figure]

**Figure S4**. Calculated steady-state (a) OH exposure, (b) [NO], (c) NO:HO$_2$, and (d) fractional loss to reaction with OH, O$_3$ and NO$_3$ as a function of input [N$_2$O] corresponding to isoprene + OH oxidation conditions at low OH exposure in the PAM reactor. Error bars represent uncertainty in model outputs (Peng et al., 2015) and in accuracy of N$_2$O flow controller.

4.  (2) NO to NO2 ratio also deviates from ambient, such that oxidation by NO3 radical becomes non-negligible (along with production of HNO3, peroxyacetyl nitrates, etc.). This, the authors note could be as high as 40% of the total oxidation by OH, O3 and NO3 combined. Perhaps the BVOC products from OH vs O3 vs NO3 can be separated using CIMS data, but the presence of different RO2 isomers resulting from each oxidant that may react with one another (RO2+RO2) may mean this simple attribution may not be possible, particularly if the products possess different functional group but same molecular composition.

**Response**: (1) It is possible to operate under conditions where NO:NO$_2$ is close to ambient ratios using this method. The new figures generated in response to Comment #1 raised by this reviewer should clarify this point: for example, NO$_2$:NO ranges from 4 to 23 over the range of conditions examined in (new) Figures 1 and 2.

(2) Please refer to our response to Comment #7 below, where we discuss in more detail the expected ion signals for acylperoxy nitrates formed from $RO_2$ + $NO_2$ reactions and organic nitrate production initiated by VOC + $NO_3$ reactions.

(3) It is possible to establish conditions where reaction rates of $RO_2$ + $RO_2$ are competitive with $RO_2$ + $NO/HO_2$ reactions, particularly if high VOC concentrations and/or low oxidant exposures are used. If this were occurring, we would expect a dampening in the effect of changing $NO:HO_2$ on the distribution of oxidation products (given that $RO_2$ + $RO_2$ would be the main termination pathway). This is a caveat that we will state in the revised manuscript.

**Revision to Section 2**. "Mixing ratios of the gas-phase precursors entering the reactor were 36 ppb for isoprene (diluted from 1000 ppm in N2, Matheson) and 15 ppb for α-pinene (diluted from 150 ppm in N2, Matheson). These mixing ratios are a factor of 3 to 10 lower than mixing ratios that are typically required to induce homogenous nucleation of condensable oxidation products in related oxidation flow reactor studies (Lambe et al., 2011b). Minimizing precursor mixing ratios also decreases the rate of $RO_2$ self-reactions relative to $RO_2$ + $HO_2$ and $RO_2$ + $NO$ reactions. This is a goal for most laboratory experiments that is not specific to the method proposed here. However, this goal takes on added importance when $RO_2$ can be formed via OH, $O_3$ and/or $NO_3$ oxidation using this method (e.g. Section 2.1)."

5. The ppm O3 levels used here also seem at odds with statements in the abstract and elsewhere in the manuscript that seem to suggest that ppm levels of O3 are bad (lines 1-5), and that this proposed technique doesn't require ppm O3 levels.

**Response.** We agree that as written this section of the manuscript appears self-contradictory. To clarify our intended message, we revised the text as follows:

**Revision to Introduction**: "A limitation of flow reactors is the need to use parts-per-million levels of $O_3$, hindering the possibility to efficiently simulate $NO_x$-dependent SOA formation pathways. […] Here, we present a new method well suited to the characterization of $NO_x$-dependent SOA formation pathways in oxidation flow reactors. By utilizing $O(^1D)$ radicals that are generated from $O_3$ photolysis, we add $N_2O$ to generate NO via the reaction $O(^1D) + N_2O \rightarrow 2NO$ with no additional method modifications."

6. In any case, have the authors attempted to run the chamber without O3? What is the highest level of NO achieved by percent levels of N2O due to direct photolysis? Given that most suburban+rural+remote regions experience highest NO levels less than 0.5 ppb, perhaps just N2O and

ambient level of O3 would suffice? This approach may not be suitable for flow reactors, but it may be for the more traditional atmospheric simulation chamber studies.

**Response.** It is not possible to run the chamber without $O_3$, because $N_2O$ does not photolyze at 254 nm. The only source of $O(^1D)$ at 254 nm is $O_3$ photolysis. Even if 185 nm radiation were added, $O_3$ would still be produced via $O_2$ photolysis.

At 30 ppb $O_3$, 1% $N_2O$, 1% $H_2O$, and $I_{254}$ = 3.2*10$^{15}$ ph cm$^{-2}$ s$^{-1}$, and 36 ppb added isoprene, 80 sec residence time, model outputs are: $OH_{exp}$ = 5*10$^9$ molec cm$^{-3}$ sec (1-2 hr of atmospheric OH oxidation), [NO] = 0.24 ppb. As the reviewer implies, this may be suitable for some studies. However, we note that the relatively high 254 nm actinic flux that is required, combined with the relatively low OH exposure that is generated, significantly increases the potential importance of unwanted photolysis (e.g. Peng et al., 2016).

**Revision**. Because the primary intended application of the method is for use in flow reactors, we do not think manuscript revisions are necessary in response to this comment.

7. The CIMS data show that, yes, as you increase NO, the level of organic nitrates increase, and levels of most organics without a nitrate decreases. This is not surprising. What is missing is that this method of NO generation by N2O in the flow reactor can demonstrate atmospherically relevant chemistry. If CIMS is the instrument of choice, the authors need to compare CIMS spectra of the flow reactor and one that was obtained from ambient atmosphere.

**Response**.

(1) We will add two panels (e) and (f) to Figure 2 in the discussion manuscript. These panels show isoprene-related ions detected in nitrate-CIMS spectra obtained during the SOAS campaign in Centreville, AL under "low-NO" (24 ppt) and "high-NO" (0.53 ppb) ambient conditions. The revised figure and text will facilitate comparison with the same isoprene oxidation products generated in the PAM reactor. Acylperoxy nitrate ion signals formed from $RO_2$ + $NO_2$ reactions (including, but not limited to, $C_{2-3}H_{3,5}NO_5$ signals representing PAN and PPN), as well as possible effects of isoprene + $NO_3$ reactions, will be incorporated and discussed.

(2) We will add one panel (c) to Figure 3. This panel shows $\alpha$-pinene-related ions detected in nitrate-CIMS spectra obtained during the same "high-NO" conditions plotted in Figure 2f. The revised figure and text will facilitate comparison with the same ion groups detected in $\alpha$-pinene photoxidation in the

PAM reactor. Acylperoxy nitrate ion signals, as well as possible effects of α-pinene + $NO_3$ reactions, will be incorporated and discussed.

(3) The x-axis of Figure 2 in the discussions paper ranged from m/Q = 160 to 380 in order to show minor signals at m/z = 351 and 367, $(HNO_3NO_3^-)C_5H_{10}O_2(NO_3)_2$ and $(HNO_3NO_3^-)C_5H_{10}O_3(NO_3)_2$. We have changed the x-axis scale to m/Q = 160 to 330 to focus on compositional changes in the bulk of the spectra.

(4) In Figure 2 of the discussions paper, we have separated $C_{4-5}H_{4-12}O_{3-8}$ compounds into $C_4H_{4,6,8}O_{4-7}$ and $C_5H_{6,8,10,12}O_{3-8}$ compounds. This highlights the observation that the C4:C5 ratio increases with added NO due to decomposition of alkoxy (RO) radicals formed from $RO_2$ + NO reactions.

Note: In our response to Reviewer #1 Comment #7, we also address parts of Reviewer #1 Comments #4, #8 and Reviewer #2 Comments #2, #6. The "Figure 3" referenced in the text below refers to a new figure introduced in response to Comment #8.

**Revision to Section 2.2**. "Mass spectra of isoprene and α-pinene gas-phase oxidation products were obtained with an Aerodyne high-resolution time-of flight mass spectrometer (Bertram et al.,2011) coupled to an atmospheric pressure interface with a nitrate ion chemical ionization source (NO3-HRToF-CIMS, hereafter abbreviated as "NO3-CIMS") […] The output of the PAM oxidation flow reactor was sampled at 10.5 Lmin−1 through a 2' length of 0.75" OD stainless steel tubing inserted directly into the rear feedthrough plate of the reactor.

Ambient $NO_3^-$ CIMS measurements were conducted during the Southern Oxidant and Aerosol Study (SOAS) at the forest site in Centreville, AL (June 1 - July 15, 2013). At this site, emissions were dominated by local biogenic volatile organic compounds (BVOC) with occasional influence from nearby anthropogenic sources (Hansen et al., 2003). The mixing of biogenic and anthropogenic emissions at the foreest site promotes the formation of organic nitrates via oxidation of BVOC in the presence of $NO_x$ (Lee et al., 2016)."

**Revision to Section 3.2**. "Ion signals corresponding to isoprene oxidation products shown in Fig. 2 were colored based on classification in ion groups containing 2-5 carbon atoms with zero ($C_4H_{4,6,8}O_{4-7}$ and $C_5H_{6,8,10,12}O_{3-8}$), one ($C_{2-3}H_{3,5}NO_5$ and $C_5H_{7,9,11}NO_{6-11}$), and two ($C_5H_{10}N_2O_{8-10}$) nitrogen atoms, where we […] also assume that nitrate functional groups are formed from $RO_2$ + NO or $RO_2$ + $NO_2$ reactions (Sect. 2.1)."

**Revision to Section 3.2.2.**

"Following addition of $N_2O$ at ∼3% mixing ratio, the $NO_3$-CIMS spectra changed significantly at low and high OH exposures (Figs. 2b, 2d, 4). The signals of $C_{4-5}H_{4-12}O_{3-8}$ oxidation products decreased, **although the $C_4H_{4,6,8}O_{4-7}$ : $C_5H_{6,8,10,12}O_{3-8}$ ratio increased, presumably due to decomposition of alkoxy (RO) radicals formed from reactions of NO with $RO_2$ radicals containing 5 carbon atoms. T**he

signals of $C_{2-3}H_{3-5}NO_5$, $C_5H_{7,9,11}NO_{6-11}$, and $C_5H_{10}N_2O_{8-10}$ oxidation products increased.

At low OH exposure, $C_{2-3}H_{3-5}NO_5$, $C_5H_{7,9,11}NO_{3-8}$, and $C_5H_{10}N_2O_{8-10}$ signals constituted 2%, 38%, and 7% of the $NO_3$-CIMS signals, respectively (Fig. 2c), assuming equal CIMS sensitivity and transmission to all detected species. The largest signal in this spectrum was m/Q = 259, $C_5H_{11}NO_7$. This compound is a second-generation oxidation product that is formed after two reactions with OH, one $RO_2$ + NO termination reaction and one $RO_2$ + $HO_2$ termination reaction (Fig. 3) (Xiong et al., 2015). A series of additional $C_5H_{7,9,11}NO_{6-11}$ ions is also detected. The signal observed at m/Q = 288, $C_5H_{10}N_2O_{8-10}$, is a second-generation oxidation product that is formed after two reactions with OH and two $RO_2$ + NO termination reactions (Fig. 3) (Xiong et al., 2015). Other ion signals associated with dinitrate species include m/Q = 304, $C_5H_{10}N_2O_9$, and m/Q = 320, $C_5H_{10}N_2O_{10}$. Related signals were detected at m/Q = 351 and 367 (not shown), which we assume represent $(HNO_3NO_3^-)C_5H_{10}N_2O_8$ and $(HNO_3NO_3)$ $C_5H_{10}N_2O_9$ because we are not aware of other feasible $(NO_3^-)C_5$ adducts at these mass-to-charge ratios.

At high OH exposure, the same $C_5H_{7,9,11}NO_{6-11}$ and $C_5H_{10}N_2O_{8-10}$ species observed at low OH exposure were detected, but at higher concentrations and at higher dinitrate:nitrate. This is presumably due to higher $NO:HO_2$ achieved at higher $I_{254}$ and fixed $[N_2O]$ (Figs.1, S1, S4-S5). $C_{2-3}H_{3,5}NO_5$ , $C_5H_{7,9,11}NO_{6-11}$, $C_5H_{10}N_2O_{8-10}$, and signals made up 0.3%, 33% and 56%, respectively, of the $NO_3$-CIMS spectrum shown in Fig. 2d, where $C_5H_{10}N_2O_8$ was the largest signal that was detected.

To demonstrate our ability to mimic atmospheric $NO_x$-dependent photochemistry, Figures 2e and f show $C_4H_{4,6,8}O_{4-7}$, $C_5H_{6,8,10,12}O_{3-8}$, $C_{2-3}H_{3,5}NO_5$, $C_5H_{7,9,11}NO_{6-11}$, and $C_5H_{10}N_2O_{8-10}$ ion signals detected in $NO_3$-CIMS spectra at the SOAS ground site in Centreville, Alabama, USA. The spectra shown were obtained on 25 Jun 2013 (0730-1100) and 4-5 Jul 2013 (1200 – 0000) which represented periods with sustained "high" and "low" NO mixing ratios of 0.53 ± 0.17 ppb and 0.024 ± 0.025 ppb, respectively, measured at the site. Figures 2a, 2c and 2e indicate that adding $N_2O$ to the reactor increases the similarity between the composition of isoprene oxidation products generated at lower photochemical age in the reactor (Figures 2a and 2c) and under "low-NO" ambient conditions (Figure 2e). Likewise, Figures 2b, 2d and 2f indicate that adding $N_2O$ to the reactor increases the similarity between the composition of isoprene oxidation products generated at higher photochemical age in the reactor (Figures 2b and d) and at "high-NO" ambient conditions (Figure 2f). We further note that $(HNO_3NO_3^-)C_5H_{10}N_2O_{8-10}$ adducts (not shown) are observed in both laboratory and ambient spectra.

3.2.2.1. Influence of acylperoxy nitrates from $RO_2$ + $NO_2$ reactions

Acylperoxy nitrates (APNs), including peroxy acetyl nitrate (PAN, $C_2H_3NO_5$) and propionyl peroxy nitrate (PPN, $C_3H_5NO_5$), are minor components (<2%) of the spectra shown in Figs. 2c-d and 2e-f. APNs are generated from reactions of aldehydic, biogenic VOC oxidation products with OH followed by $RO_2$ + $NO_2$ termination reactions (e.g. LaFranchi et al., 2009). A comparison of Figs. 2c and 2e suggests that yields of PAN and PPN are not enhanced in the reactor compared to atmospheric conditions.

Additional APNs may be generated following the OH oxidation of methacrolein, a first-generation isoprene oxidation product. Methacryloyl peroxy nitrate (MPAN, $C_4H_5NO_5$) is a second-generation oxidation product formed after one methacrolein + OH reaction and one $RO_2$ + $NO_2$ termination reaction (Orlando et al., 1999). C4-hydroxynitrate-PAN ($C_4H_6N_2O_9$) is a third-generation oxidation product formed through the methacrolein channel after three reactions with OH, two $RO_2$ + NO termination reactions and one $RO_2$ + $NO_2$ termination reaction (Surratt et al., 2010).

Neither $C_4H_5NO_5$ nor $C_4H_6N_2O_9$ were detected in the laboratory and ambient $NO_3$-CIMS spectra shown in Figs. 2c and 2d. Either these compounds were oxidized or thermally decomposed prior to detection, or their signals were below detection limit. $C_4H_7NO_5$, which is formed after one methacrolein + OH reaction and one $RO_2$ + NO termination reaction (Surratt et al., 2010), was detected (Fig. 3). Taken together, these observations suggest that yields of APNs are not significantly enhanced in the reactor compared to atmospheric conditions.

3.2.2.2. Influence of isoprene + $NO_3$ reactions

Based on the calculated isoprene + OH and isoprene + $NO_3$ reaction rates (Figs. S4-S5) we assume that isoprene + $NO_3$ reactions have a minor influence on the $NO_3$-CIMS spectra shown in Figs. 2c and 2d. This assumption is further supported by the similarity between laboratory and ambient $NO_3$-CIMS spectra, the latter of which were obtained during the daytime and thus with minimal $NO_3$ exposure (Figs. 2e and f). Specific operating conditions different than those used in this study could increase the relative influence of isoprene + $NO_3$ reactions. In this hypothetical situation, enhanced yields of $C_5H_7NO_5$, $C_5H_8N_2O_8$ and $C_5H_{10}N_2O_8$ might occur following two reactions with $NO_3$ (Rollins et al., 2009). In addition, $C_5H_{10}N_2O_9$ may be generated from one isoprene + $NO_3$ reaction followed by one $RO_2$ + $HO_2$ termination reaction (Schwantes et al., 2015).

All four of these ions are detected in the spectra shown in Fig. 2, although $C_5H_8N_2O_8$ (not shown in Fig. 2) is present at 0.5% of the intensity of $C_5H_{10}N_2O_8$. If $C_5H_8N_2O_8$ : $C_5H_{10}N_2O_8$ is significantly different under $NO_3$-dominated conditions, this ratio could distinguish the relative rates of isoprene + OH and isoprene + $NO_3$ reactions. Otherwise, it is not clear that the expected product distributions are

significantly different whether isoprene is oxidized by OH or $NO_3$ in the presence of $NO_x$.

**Revisions to Section 3.3**. "Figure 3 shows $NO_3$-CIMS mass spectra of products generated from the oxidation of α-pinene (C10H16). […] $C_{2-3}H_{3,5}NO_5$, $C_5H_7NO_{6-11}$ […] containing one nitrogen atom […] As was the case with isoprene oxidation products, we assumed nitrogen atoms present in α-pinene oxidation products were associated with nitrate functional groups formed from $RO_2$ + NO or $RO_2$ + $NO_2$ reactions.

**Revisions to Section 3.3.1**. "the signal detected at "low-NOx" conditions […] comprised 5%, 36%, 46%, and 4%, respectively, again assuming equal CIMS sensitivity and transmission to all detected species. The C10 monomers and C19−20 dimers compounds that were observed are often associated with atmospheric new particle formation events […] The remaining ~10% of the signal was classified into $C_{2-3}H_{3,5}NO_5$, […] and $C_{10}H_{15,17}NO_{4-14}$ ion groups."

**Revision to Section 3.3.2**. "As was the case with $NO_3$-CIMS spectra of isoprene oxidation products, the addition of $N_2O$ to the reactor significantly changed the mass spectrum of α-pinene oxidation products (Figs. 3b and 5). At [$N_2O$] = 3.2%, organic nitrates and dinitrates comprised 62% of the total ion signal (Fig. 3b inset). […] The largest organic nitrate signals in this spectrum were at m/Q = 329, $C_8H_{13}NO_9$, followed by $C_{10}H_{15}NO_9$ (m/Q = 355), $C_{10}H_{16}N_2O_9$ (m/Q = 354), and $C_{10}H_{15}NO_8$ (m/Q = 339).

Figure 3c shows $C_5H_6O_{5-7}$, $C_{6-9}H_{8,10,12,14}O_{6-12}$, $C_{10}H_{14,16,18}O_{5-14}$, $C_{19-20}H_{28,30,32}O_{9-18}$, $C_{2-3}H_{3,5}NO_5$, $C_5H_7NO_{6-11}$, $C_{6-9}H_{9,11,13,15}NO_{5-10}$, $C_{10}H_{15,17}NO_{4-14}$, and $C_{10}H_{16,18}N_2O_{6-13}$ signals detected with $NO_3$-CIMS spectra at the Centreville site. The spectra shown here were obtained during the sampling period shown in Fig. 2f and, given the large number of compounds, may include contributions from HOM precursors other than α-pinene. A comparison of Figs. 3a-3c indicates that adding $N_2O$ to the reactor increases the similarity between the composition of α-pinene oxidation products generated in the reactor and under "high-NO" ambient conditions, especially in regards to the enhanced $C_5H_6O_{5-7}$, $C_{6-9}H_{9,11,13,15}NO_{5-10}$, $C_{10}H_{15,17}NO_{4-14}$, and $C_{10}H_{16,18}N_2O_{6-13}$ signals. $C_5H_6O_7$, $C_{10}H_{15,17}NO_{9-14}$ and $C_{10}H_{16,18}N_2O_{9-13}$ signals are higher in Fig. 3b than in Fig. 3c.

3.3.2.1. Detection of acylperoxy nitrates (APN) from $RO_2$ + $NO_2$ reactions
We examined $NO_3$-CIMS spectra of α-pinene oxidation products for the presence of PAN and PPN as components of the $C_{2-3}H_{3,5}NO_5$ group. Figs. 3b and 3c indicate that PAN (m/Q = 183, $C_2H_3NO_5$) and PPN (m/Q = 197, $C_3H_5NO_5$) are formed at lower yields (<0.4%) than were observed with isoprene (Fig. 2c and 2d). Thus, results suggest that yields of PAN and PPN from reaction of α-pinene-derived-$RO_2$ with $NO_2$ are not enhanced in the reactor compared to atmospheric conditions.

$C_9H_{13}NO_6$ and $C_{10}H_{15}NO_{6-8}$ are APNs generated following OH oxidation of pinonaldehyde, a major first-generation oxidation product of $\alpha$-pinene, with termination by $RO_2 + NO_2$ reaction (e.g. Eddingsaas et al., 2012). All four compounds are detected in the reactor and ambient $NO_3$-CIMS spectra shown in Figs. 3b and 3c, with $C_{10}H_{15}NO_{6-8}$ signals among the largest in the spectra. If these signals represent APNs, they appear to be important in both laboratory and atmospheric conditions.

3.3.2.2. Influence of $\alpha$-pinene + $NO_3$ reactions

Our calculations suggest that $\alpha$-pinene + $NO_3$ reactions may compete with $\alpha$-pinene + OH reactions at the experimental conditions used to generate the $NO_3^-$-CIMS spectrum shown in Fig. 3b. If this were the case, enhanced yields of $C_{10}H_{15}NO_6$ are anticipated from $\alpha$-pinene + $NO_3$ reaction to generate pinonaldehyde, followed by pinonaldehyde + $NO_3$ reaction and $RO_2 + NO_2$ termination (Perraud et al., 2010; Nah et al., 2016). Other minor $\alpha$-pinene + $NO_3$ products detected with CIMS include $C_{10}H_{15}NO_5$, $C_9H_{13}NO_6$, $C_{10}H_{16}N_2O_7$, and $C_{10}H_{15}NO_9$ (Nah et al., 2016). We hypothesize that if $\alpha$-pinene + $NO_3$ reactions influence the spectrum shown in Fig. 3b, $C_{10}H_{15}NO_6 : C_{10}H_{15}NO_8$ should be higher in Fig. 3b than in Fig. 3c. Instead, the $C_{10}H_{15}NO_6 : C_{10}H_{15}NO_8$ ratio was 0.12 in the reactor (Fig. 3b) and 0.28 at the Centreville site (Fig. 3c) during a daytime period with negligible $NO_3$ influence.

Dinitrates ($C_{10}H_{16,18}N_2O_{6-13}$) shown in Fig. 3b may originate from two $\alpha$-pinene + OH reactions followed by two $RO_2$ + NO terminations, or one $\alpha$-pinene + $NO_3$ reaction followed by one $RO_2$ + NO termination. Given comparable OH and $NO_3$ reaction rates under these conditions (Fig. S6e), we hypothesize that the majority of dinitrate signals should originate from $\alpha$-pinene + $NO_3$ reactions if their yields are not oxidant-dependent. If this is the case, $C_{10}H_{16,18}N_2O_{6-13} : C_{10}H_{15,17}NO_{4-14}$ should be larger in Fig. 3b than in Fig. 3c. However, $C_{10}H_{16,18}N_2O_{6-13} : C_{10}H_{15,17}NO_{4-14}$ was 0.23 in the spectrum shown in Fig. 3b and 0.61 in the spectrum shown in Fig. 3c.

Thus, while the calculated $\alpha$-pinene + $NO_3$ oxidation rate is significant (Fig. S6e), it is not clear that $\alpha$-pinene + $NO_3$ oxidation products significantly affect the spectrum shown in Fig. 3b. This may be due to significantly lower organic nitrate yields from $\alpha$-pinene + $NO_3$ than from $\alpha$-pinene + OH reactions in the presence of NO (Fry et al., 2014; Rindelaub et al., 2015).

**Revisions to Figures 2 and 3.**

[Figure]

**Figure 2.** NO$_3$-CIMS mass spectra of isoprene oxidation products generated at [H$_2$O] = 1%, [O$_3$] = 5 ppm, mean residence time = 80 sec: (a) I$_{254}$ = 6.4×10$^{13}$ ph cm$^{-2}$ sec$^{-1}$, [N$_2$O] = 0%; (b) I$_{254}$ = 3.2×10$^{15}$ ph cm$^{-2}$ sec$^{-1}$, [N$_2$O] = 0 %; (c) I$_{254}$ = 6.4×10$^{13}$ ph cm$^{-2}$ sec$^{-1}$, [N$_2$O] = 3.2 %; (d) I$_{254}$ = 3.2×10$^{15}$ ph cm$^{-2}$ se$^{c-1}$, [N$_2$O] = 2.9 %; (e) and (f) C$_4$H$_{4,6,8}$,O$_{4-7}$, C$_5$H$_{8,10,12}$O$_{3-8}$, C$_{2-3}$H$_{3,5}$NO$_5$ , C$_5$H$_{7,9,11}$NO$_{6-11}$, and C$_5$H$_{10}$N$_2$O$_{8-10}$ ion groups detected at the SOAS ground site in Centreville, Alabama, USA during (e) "low-NO" and (f) "high-NO" conditions (see text for additional details; C$_5$H$_6$O$_{5-7}$ ions removed from SOAS spectra due to larger contributions from $\alpha$-pinene + OH oxidation products (Fig. 3). "C$_x$" or "O$_x$" indicate number of carbon or oxygen atoms in labeled ions (not including oxygen atoms associated with nitrate functional groups).

[Figure]

**Figure 3.** $NO_3^-$ -CIMS mass spectra of α-pinene oxidation products generated at $[H_2O]$ = 0.07%, $[O_3]$ = 5 ppm, mean residence time = 80 sec: (a) $I_{254}$ = 2.8×10$^{15}$ ph cm$^{-2}$ se$^{c-1}$, $[N_2O]$ = 0 %; (b) $I_{254}$ = 2.8×10$^{15}$ ph cm$^{-2}$sec$^{-1}$, $[N_2O]$ = 3.2 %. (c) $C_5H_{6,8}O_{5-7}$, $C_{6-9}H_{8,10,12,14}O_{6-12}$, $C_{10}H_{14,16,18}O_{5-14}$, $C_{19-20}H_{28-32}O_{9-18}$, $C_{2-3}H_{3,5}NO_5$, $C_5H_7NO_{6-11}$, $C_{6-9}H_{9,11,13,15}NO_{5-10}$, $C_{10}H_{15,17}NO_{4-14}$, and $C_{10}H_{16,18}N_2O_{6-13}$ ion groups detected at the SOAS ground site in Centreville, Alabama, USA during "high-NO" conditions shown in Fig. 2f (note: $C_5H_7NO_{6-11}$ signals in SOAS spectra also contributed from isoprene + OH oxidation products). "$C_x$" or "$O_x$" **Figure 2.** $NO_3^-$ -CIMS mass spectra of α-pinene oxidation products generated at $[H_2O]$ = 0.07%, $[O_3]$ = 5 ppm, mean residence time = 80 sec: (a) $I_{254}$ = 2.8×10$^{15}$ ph cm$^{-2}$ se$^{c-1}$, $[N_2O]$ = 0 %; (b) $I_{254}$ = 2.8×10$^{15}$ ph cm$^{-2}$sec$^{-1}$, $[N_2O]$ = 3.2 %. (c) $C_5H_{6,8}O_{5-7}$, $C_{6-9}H_{8,10,12,14}O_{6-12}$,

**Revisions to References. We added the following citations to references:**

Eddingsaas, N. C., Loza, C. L., Yee, L. D., Seinfeld, J. H., and Wennberg, P. O.: α-pinene photooxidation under controlled chemical conditions – Part 1: Gas-phase composition in low- and high-NO$_x$ environments, Atmos. Chem. Phys., 12, 6489-6504, doi:10.5194/acp-12-6489-2012, 2012.

J. L. Fry, D. C. Draper, K. C. Barsanti, J. N. Smith, J. Ortega, P. M. Winkler, M. J. Lawler, S. S. Brown, P. M. Edwards, R. C. Cohen, and L. Lee. Secondary Organic Aerosol Formation and Organic Nitrate Yield from NO3 Oxidation of Biogenic Hydrocarbons. Environ. Sci. & Technol., *48* (20), 11944-11953, 2014.

Hansen, D. A., Edgerton, E. S., Hartsell, B. E., Jansen, J. J., Kandasamy, N., Hidy, G. M., & Blanchard, C. L. The Southeastern Aerosol Research and Characterization Study: Part 1--Overview. Journal of the Air & Waste Management Association, *53*(12), 1460-1471, 2003.

LaFranchi, B. W., Wolfe, G. M., Thornton, J. A., Harrold, S. A., Browne, E. C., Min, K. E., Wooldridge, P. J., Gilman, J. B., Kuster, W. C., Goldan, P. D., de Gouw, J. A., McKay, M., Goldstein, A. H., Ren, X., Mao, J., and Cohen, R. C.: Closing the peroxy acetyl nitrate budget: observations of acyl peroxy nitrates (PAN, PPN, and MPAN) during BEARPEX 2007, Atmos. Chem. Phys., 9, 7623-7641, 2009.

Orlando, J. J., Tyndall, G. S., and Paulson, S. E. Mechanism of the OH-initiated oxidation of methacrolein. Geophys. Res. Lett., 26(14), 2191 – 2194, 1999.

Nah, T., Sanchez, J., Boyd, C. M., and N. L. Ng, N. L. Photochemical Aging of alpha-pinene and beta-pinene Secondary Organic Aerosol formed from Nitrate Radical Oxidation, Environ. Sci. Technol., 50, 222-231, 2016.

Perraud, V., Bruns, E. A. , Ezell, M. J. , Johnson, S. N., Greaves, J., and Finlayson-Pitts, B. J.. Identification of Organic Nitrates in the $NO_3$ Radical Initiated Oxidation of α-Pinene by Atmospheric Pressure Chemical Ionization Mass Spectrometry. Environ. Sci. Technol., 44 (15), 5887-5893, 2010.

Rindelaub, J. D., McAvey, K. M. and Shepson, P. B. The photochemical production of organic nitrates from α-pinene and loss via acid-dependent particle phase hydrolysis. Atmos. Environ., 100, 193–201, 2015.

Rollins, A. W., Kiendler-Scharr, A., Fry, J. L., Brauers, T., Brown, S. S., Dorn, H.-P., Dubé, W. P., Fuchs, H., Mensah, A., Mentel, T. F., Rohrer, F., Tillmann, R., Wegener, R., Wooldridge, P. J., and Cohen, R. C.: Isoprene oxidation by nitrate radical: alkyl nitrate and secondary organic aerosol yields, Atmos. Chem. Phys., 9, 6685-6703, doi:10.5194/acp-9-6685-2009, 2009.

Schwantes, R. H., Teng, A. P., Nguyen, T. B., Coggon, M. M., Crounse, J. D., St. Clair, J. M., Zhang, X., Schilling, K. A., Seinfeld, J. H., and Wennberg, P. O. Isoprene $NO_3$ Oxidation Products from the $RO_2$ + $HO_2$ Pathway, J. Phys. Chem. A, 119 (40), 10158-10171, 2015.

Surratt, J. D.; Chan, A. W. H.; Eddingsaas, N. C.; Chan, M. N.; Loza, C. L.; Kwan, A. J.; Hersey, S. P.; Flagan, R. C.; Wennberg, P. O.; Seinfeld, J. H. Reactive Intermediates Revealed in Secondary Organic Aerosol Formation from Isoprene. Proc. Nat. Acad. Sci., 107 (15), 6640–6645, 2010.

8. The discussion sections on the types of oxidation products observed are less than convincing, lacking the detailed mechanism discussions typically included in such studies. As such, these sections read more like speculation.

**Response.** Since the reviewer did not specify specific text that reads as speculative, we assume (s)he is referring to the following text in Sections 3.2.1 and 3.2.2:

**P7, L10-L12**: "The signal at m/Q = 230, $C_5H_{12}O_6$ [...] is likely a second-generation oxidation product that contains two hydroxyl (OH) and two peroxide (OOH) functional groups (Krechmer et al., 2015; St Clair et al., 2016)."

**P7, L15-L16**: "Previously-identified multi-generation isoprene oxidation products such as $C_5H_{10}O_5$, $C_5H_{12}O_5$, and $C_5H_{10}O_6$ (Surratt et al., 2006; Krechmer et al., 2015; Liu et al., 2016; St Clair et al., 2016) were also detected at significant intensity under low-$NO_x$ conditions."

**P7, L19-L22**: "At high OH exposure, $C_5H_{12}O_7$ was the second-largest peak in the spectrum. These highly oxygenated isoprene oxidation products are likely also important in SOA formation processes. We note that $C_5H_{10}O_7$ is a proposed third-generation, tri-hydroperoxycarbonyl product of isoprene + OH in the absence of $NO_x$ (Peeters et al., 2014)."

**P7-P8, L32-L1**: "The largest signal in this spectrum was m/Q = 259, $C_5H_{11}O_4NO_3$. This compound is likely a second-generation oxidation product that contains two hydroxyl functional groups and one nitrate functional group (Xiong et al., 2015; Liu et al., 2016). A series of additional $C_5H_{9,11}O_{3-8}NO_3$ ions is also detected. The signal observed at m/Q = 288, $C_5H_{10}O_2(NO_3)_2$, is likely a second-generation oxidation product that contains two hydroxyl and two nitrate functional groups (Xiong et al., 2015; Liu et al., 2016)."

To address this comment, we will add a new figure summarizing known pathways of the isoprene + OH oxidation mechanism that form the major ions that are detected with $NO_3$-CIMS (Figure 5 below; "Figure 3" in revised text. Additional benefits of this figure are that it will introduce the proposed structures shown in Figure 4 and will illustrate the steps that are necessary to form methacryloyl peroxy acetyl nitrate (MPAN) and C4-hydroxynitrate-PAN from $RO_2$ + $NO_2$ termination pathways (see Comment #2 by Reviewer 2).

**Revision to Section 3.2.** "Figure 2 shows $NO_3^-$-CIMS mass spectra of products generated from the oxidation of isoprene [...] Thus, it is unlikely that OH suppression at "high OH" and "high $NO_x$" significantly affected the $NO_3^-$-CIMS spectra shown in Fig. 2.

==To aid interpretation of results shown in Fig. 2, Fig. 3 summarizes several known isoprene + OH reaction pathways that are terminated by reactions of $RO_2$ with $HO_2$, NO, or $NO_2$. As will be discussed in the following sections, these==

pathways yield multigenerational oxidation products with chemical formulas corresponding to several of the major ions that are plotted in Fig. 2."

**Revision to Section 3.2.1.** " […] The signal at $m/Q = 230$, $C_5H_{12}O_6$ ($NO_3^-$) omitted for brevity here and elsewhere), was the largest signal detected at both low and high OH exposures at "low-NOx" conditions. This species is  a second-generation oxidation product generated from two reactions with OH and two $RO_2$ + $HO_2$ termination reactions (Fig. 3) (Krechmer et al., 2015; St Clair et al., 2016) and is typically associated with isoprene SOA formation and growth under "low-NOx" conditions (Liu et al., 2016). […] Previously-identified multi-generation isoprene oxidation products such as $C_5H_{10}O_5$, $C_5H_{12}O_5$, and $C_5H_{10}O_6$ (Surratt et al., 2006; Krechmer et al., 2015; St Clair et al., 2016) were also detected at significant intensity under low-NOx conditions. These species are formed after two reactions with OH, one $RO_2$+ $HO_2$ termination reaction and one $RO_2$ + $RO_2$ termination reaction (Fig. 3). […] At […] We note that $C_5H_{10}O_7$ is a proposed third-generation, tri-hydroperoxycarbonyl product formed after one reaction with OH, two hydrogen shifts and one $RO_2$ + $HO_2$ termination reaction as shown in Fig. 3 (Peeters et al., 2014).

**Revision to Section 3.2.2.** Please see our response to Comment #7 raised by this reviewer. In that response, we revised additional text to incorporate changes associated with the response to this comment.

**Revision to Reference**. The following citation will be added:

J. Liu, E. L. D'Ambro, B. H. Lee, F. D. Lopez-Hilfiker, R. A. Zaveri, J. C. Rivera-Rios, F. N. Keutsch, S. Iyer, T. Kurten, Z. Zhang, A. Gold, J. D. Surratt, J. E. Shilling, and J. A. Thornton. Efficient Isoprene Secondary Organic Aerosol Formation from a Non-IEPOX Pathway. Environ. Sci. Technol., *50* (18), 9872-9880, 2016.

9. Does the model account for RO2 chemistry? Is there a model output for the various organic molecular compositions observed or at least groups of organics (i.e. Krechmer 2015 ES&T)?

**Response**. Yes, the model does account for simplified $RO_2$ chemistry. We revised the manuscript to indicate this.

**Revision to Section 2.3**. "The model also includes simplified $RO_2$ chemistry, which is incorporated using the reactions listed below. The addition of these reactions constrain the effects of added isoprene or $\alpha$-pinene (species "X" below) on steady-state [OH], [$HO_2$] and [NO]:

OH + X --> RO2 + H2O
RO2 + NO -> RO + NO2                      k = 8e-12
RO2 + HO2 -> ROOH + O2                    k = 1.2e-11
ROOH + OH -> RO2 + H2O            k_ = 5.3e-12*exp(190./T)*0.6
ROOH + OH -> R'HO + OH + H2O       k = 5.3e-12*exp(190./T)*0.4
RO2 + OH -> RPO2 + H2O                    k = 2.3e-10
RO2 + RO2 -> ROOR                        k = 5e-12
RO + O2 -> RPO + HO2                       k = 6e-15;
RO2 + NO + M --> RNO3 + M          k = 0.02*k_ro2_no
RO + NO + M -> RONO + M            k = 3e-11
RO + NO2 + M -> RONO2 + M                 k = 3e-11

Calculated OH exposures…"

10.       How much of NO:HO2 changes (x-axis; figures 4 and 5) are due to the reaction of NO with HO2? Is RO2 accounted for in the calculation of NO and HO2?

**Response.** $RO_2$ is (crudely) accounted for in the calculation of NO and $HO_2$ (see reply to previous Comment #9). We calculated the relative rates of NO + $HO_2$, NO + $RO_2$ and $HO_2$ + $RO_2$ reactions for an experiment in which 36 ppb isoprene was

added to the reactor. We specified the following model input parameters: mean residence time = 80 sec, $I_{254}$ = 3.2*10$^{15}$ ph cm$^{-2}$ sec, [$O_3$] = 5 ppm, [$H_2O$] = 1%, and [$N_2O$] = 3%. The maximum [$RO_2$] calculated by the model was 2.6 ppb. At this timestep, [NO] = 5.3 ppb and [$HO_2$] = 1.4 ppb. Thus,

$k[RO_2][NO]$ = 6.7*10$^{10}$ molec cm$^{-3}$ sec
$k[RO_2][HO_2]$ = 2.6*10$^{10}$ molec cm$^{-3}$ sec
$k[NO][HO_2]$ = 3.9*10$^{10}$ molec cm$^{-3}$ sec

This calculation implies that the rate of NO + $HO_2$ reactions is comparable to, or greater than, $RO_2$ + NO and $RO_2$ + $HO_2$ reactions.

**Revision to Section 3.4.** "Figures 4 and 5 show normalized signals of the representative groups of isoprene and α-pinene oxidation products as a function of increasing NO:$HO_2$, which may be influenced by NO + $HO_2$, NO + $RO_2$ and $HO_2$ + $RO_2$ reactions in the reactor that are accounted for in the model. For each group of compounds, signals obtained at a specific NO:$HO_2$ were normalized to the maximum observed signal. NO:$HO_2$ is correlated with the relative branching ratios of $RO_2$ + $HO_2$ and $RO_2$ + NO reactions that govern the distribution of oxidation products observed in Figs. 2 and 3."

11.      Figure 2, judging from the y-axes, much higher signal levels are observed at higher I254. Is this the result of production of later-generation oxidation products? Or just more complete oxidation? Was the amount of parent BVOC oxidized measured?

**Response**. We refer the reviewer to the following text in the discussions manuscript; to further clarify this point, we revised the text as shown below. We did not measure the amount of parent BVOC that was oxidized.

Section 3.2.1, P7, L13-L15: "Signals in Figs. 2c-d are approximately 10 times higher than in Figs. 2a-b because additional OH exposure produces higher yields of multi-generation oxidation products that are detected with NO3-CIMS."

**Revision to Section 3.2.1**: "As shown in Figs.S4 and S5, corresponding OH exposures ranged from (1.7–2.0)×10$^{10}$ (Fig. 2a and 2c; calculated >82% of isoprene reacted) and (0.52 –2.1)×10$^{12}$ molec cm$^{-3}$ sec (Fig. 2b and 2d; calculated ~100% of isoprene reacted), respectively".

12.      Figure 4 is misleading. From what I understand, the CIMS identifies molecular compositions but cannot assign structure/isomer/functional groups. What is the source of the drawings on top of figure 4? How were they determined?

**Response**. We revised the Figure 4 caption to indicate that these are suggested structures based on consistency with previous work. Please also see our response to Comment #8 from this reviewer, where we present a mechanistic figure explaining the anticipated reaction pathways to form these compounds. We will also replace "$C_wH_xO_y(NO_3)_z$" with "$C_wH_xN_zO_{y+3z}$" representation where applicable throughout the manuscript.

**Revision to Figure 4 caption.** "Normalized signals of $C_{4-5}H_{4-12}O_{3-8}$, $C_5H_{7,9,11}NO_{6-11}$, and $C_5H_{10}N_2O_{8-10}$ isoprene oxidation products as a function of modeled $NO:HO_2$. For each of the species classes, signals were normalized to the maximum signal. Proposed structures for $C_5H_{12}O_6$, $C_5H_{11}NO_7$, and $C_5H_{10}N_2O_8$ signals are shown as representative ions for each species class (St. Clair et al., 2016; Xiong et al., 2015)."

**Reviewer #2**

1. While the main goal of this paper is an experimental demonstration of proof of concept of oxidation flow reactors with a dominant contribution of the RO2+NO pathway, it failed to provide convincing evidence that the observed N-containing product formation is due to that pathway. The authors have not ruled out peroxynitrate formation (see comment #2) or products from NO3 oxidation. Although the authors tried explaining the trend of N-containing product signals vs. NO:HO2 in Section 3.4 and Figures 4 and 5, their discussion seemed to start with the assumption that most N-containing products are formed by RO2+NO, i.e. organic nitrates. This assumption needs to be demonstrated. To clearly demonstrate organic nitrate formation through RO2+NO, I suggest conducting additional experiments where short-chain alkanes (e.g. butane) are used as precursor and OH exposure is limited. In these experiments, acyl RO2 formation is limited, NO3 addition is impossible, RO2 autoxidation is unlikely, and CIMS data should be much simpler to analyze and could contain much clearer information pointing to organic nitrate formation.

**Response**. It is a fair point that the assumption of N-containing products being formed from $RO_2$ + NO reactions could have been justified more rigorously. In attempt to incorporate this comment into our revised manuscript, we made significant revisions (e.g our response to Comments #7 and #8 raised by Reviewer #1). This reviewer's suggestion to conduct experiments with short-chain alkanes is well taken but in our opinion is not necessary after the aforementioned revisions. Further, we think it is not practical in conjunction with use of $NO_3$-CIMS as the detector which is more sensitive to multifunctional, highly oxidized,

multigenerational oxidation products that are unlikely to be generated in the experimental systems suggested by the reviewer.

**Revision**. Please see our response to the comments below.

> 2. This study completely ignored the possibility of peroxynitrate formation in the PAM. Acylperoxynitrates are relatively stable, especially at short residence times such as in the PAM. NO3-CIMS is unable to distinguish organic nitrates from acylperoxynitrates. Showing tentative structure attribution in e.g. Figure 4 without emphasizing the caveats is very misleading. If N-containing products are mostly acylperoxynitrates formed by RO2+NO2, the new method in this paper may not be as useful as claimed. It should be demonstrated that acylperoxynitrates are not dominant products in the experiments shown in this paper.

**Response**. We examined the $NO_3$-CIMS spectra obtained under laboratory conditions (with added $N_2O$) and ambient conditions (see Comment #7 by Reviewer 1) for the presence of the following acylperoxy nitrates (APNs):

- Peroxy acetyl nitrate (PAN) at m/Q = 183, $(NO_3^-)C_2H_3NO_5$
- Peroxy propionyl nitrate (PPN) at m/Q = 197, $(NO_3^-)C_3H_5NO_5$

(isoprene only, as proposed by Surratt et al., 2010)
- Methacryloyl peroxy nitrate (MPAN) at m/Q = 209, $(NO_3^-)C_4H_5NO_5$
- C4-hydroxynitrate-PAN at m/Q = 288, $(NO_3^-)C_4H_6N_2O_9$

($\alpha$-pinene only, as proposed by Eddingsaas et al., 2012)
- Norpinonaldehyde PAN at m/Q = 293, $(NO_3^-)C_9H_{13}NO_6$
- Pinonaldehyde PAN at m/Q = 307, $(NO_3^-)C_{10}H_{15}NO_6$
- Unidentified PAN at m/Q = 323, $(NO_3^-)C_{10}H_{15}NO_7$
- Unidentified PAN at m/Q = 339, $(NO_3^-)C_{10}H_{15}NO_8$

If the reviewer is aware of other known APNs derived from isoprene or $\alpha$-pinene, we would be happy to add them to the revised discussion.

**Revision.** Please see our response to Comment #7 raised by Reviewer #1, where we also incorporated our response to Comment #2 raised by this reviewer. Overall, we conclude:
(1) isoprene-derived APN's are formed in at most minor yields in the reactor through the methacrolein channel, because neither MPAN or C4-hydroxynitrate-PAN are detected

(2) A subset of the $\alpha$-pinene-derived organic nitrates may represent APNs formed through the pinonaldehyde channel. If that is the case, the measurements imply that these species are important in both laboratory and ambient conditions.

3. This paper claimed that high NO concentrations and a variable NO:HO2 were obtained by the new method. But these quantities were all only calculated by a model. If these quantities are measured, the claims would be much more convincing, especially for a study that is mainly an experimental demonstration of proof of concept. NO is easy to measure and a method for measuring HO2 with CIMS has recently been published (Sanchez et al., 2016).

**Response**. Please see our response to Comment #1 raised by Reviewer 1.

**Revision**. We made significant revisions to the text and added 3 new figures in response to Comment #1 raised by Reviewer 1, which is similar to this comment.

4. At the highest UV used in this study (3.2x10^15 ph cm-2 sec at 254 nm), the UV light can be estimated to destroy >90% O3 by the end of a residence time of 80 s. Moreover, NO, HO2, OH etc. can also consume O3. As a result, both OH and NO production, sustained by O3, would be greatly reduced then and the chemistry close to the exit of the reactor would remarkably deviate from the authors' original design. However, most experiments in this study were conducted at (nearly) the highest UV. The authors should clarify the impact of O3 being largely destroyed on the chemistry in the PAM.

**Response.** It is true that processes in the reactor consume $O_3$, but the reviewer's assertion that >90% of $O_3$ is destroyed at $I_{254}$ = 3.2x10$^{15}$ ph cm$^{-2}$ sec$^{-1}$ is not correct. A significant fraction of O($^1$D) formed from $O_3$ photolysis is quenched to O($^3$P) following collisional stabilization with $O_2$ or $N_2$. O($^3$P) then recombines with $O_2$ to regenerate most of the $O_3$ that is photolyzed. For example, at the conditions mentioned above, if [$O_3$]$_{initial}$ = 5 ppm:

[$H_2O$] = 0.07%, [$N_2O$] = 0%: [$O_3$]$_{final}$ = 4.8 ppm (4% ozone destruction)
[$H_2O$] = 1%, [$N_2O$] = 0%: [$O_3$]$_{final}$ = 3.4 ppm (32% destruction, upper limit to "low-NO" studies presented in this paper)
[$H_2O$] = 1%, [$N_2O$] = 5%: [$O_3$]$_{final}$ = 1.6 ppm (68% destruction, upper limit to "high-NO" studies presented in this paper)
The more relevant point is that less $O_3$ is regenerated in the presence of additional O($^1$D) sinks such as $H_2O$ and $N_2O$. It is not clear that this significantly changes the main conclusions of this paper.

**Revision to Section 2.1**. "Using $N_2O$ as the $NO_x$ precursor has the following advantages over the simple addition of NO to the carrier gas. First, due to continuous production of $O(^1D)$ from $O_3$ photolysis inside the reactor (along with minor consumption of $N_2O$), the spatial distribution of NO and $NO_2$ is more homogenous. Second, attainable steady-state mixing ratios of NO from $N_2O$ + $O(^1D)$ reactions (ppb levels) are orders of magnitude higher than simple NO injection (sub-ppt levels) as inferred from photochemical model simulations described below in Sect. 2.3. Gradients in $[O(^1D)]$ due to its reaction with $H_2O$ and $N_2O$ alter the spatial distribution of $O_x$, $HO_x$ and $NO_x$ in the reactor. To first order, gradients in $[O(^1D)]$ should decrease both $[HO_2]$ and [NO] to a similar extent, and therefore the relative rates of $RO_2$ + $HO_2$ and $RO_2$ + NO termination pathways should remain the same."

5. In the modeling of this study, it is unclear whether the effect of VOCs in affecting the OH concentration (by shortening OH lifetime) has been considered. If it has already been considered, this should be clarified.

**Response**. Yes, this has been considered. Please see our response to Comment #9 raised by Reviewer 1.

**Revision**. Please see our response to Comment #9 raised by Reviewer 1.

6. Organic ozonolysis and NO3 reactions may be important compared to VOC reactions with OH in the PAM. In particular, a-pinene can be consumed by as much as 40% by NO3 as mentioned in the paper. The authors should rule out interferences due to NO3 reactions in the observed MS spectra and discuss the importance of organic ozonolysis relative to reactions with OH.

**Response.** Please see our replies to Comments #3 and #7 by Reviewer #1. In response to Comment #3 by the other reviewer, we modified Figures S4d, S5d and S6e in the Supplement to indicate the relative roles of OH, $O_3$ and $NO_3$ for oxidation of isoprene or $\alpha$-pinene under the experimental conditions that are used. This axis replaced the "OH:NO3" axis shown previously in Figs. S4d, S5d and S6e. In response to Comment #7 by the other reviewer, we also incorporated our response to this comment, where we added subsections discussing potential interferences from isoprene + $NO_3$ and $\alpha$-pinene + $NO_3$ reactions. The revised Figures S4-S6 imply that the rate of ozonolysis reactions is not fast enough to compete with the rates of OH or $NO_3$ reactions.

7. Page 5, Line 16: this reaction leads to total N in the model being unconserved. It is unclear to me if this reaction plays a major role. If not, it

should be stated for clarity; otherwise, total N-containing species could be significantly underestimated by the model and a correction to this problem would be needed.

**Response**. Thank you for pointing this out. $HNO_4$, which would be detected at m/Q = 141, $(NO_3^-)HNO_4$, is below the NO3-CIMS detection limit in these measurements; thus, we assume it is does not play a major role.

**Revision to Section 2.3.** We removed R16 from the list of equations.

8. Page 6, Line 15: it is stated that at lower [N2O], increasing [O3] increases [NO]. But this is not clear, since both NO production and loss are approximately proportional to [O3] and change in [O3] would have little effect. If the authors did observe this, they need to provide more detail and explain its cause better.

**Response**. Thank you for your insight, which helped us catch a mistake in the model that becomes important at conditions using 50 ppm $O_3$. After rerunning the model with 0.5, 5, and 50 ppm input $O_3$ and 1% input $N_2O$, the trends change: the maximum [NO] occurs at 5 ppm input $O_3$. This is because at 0.5 ppm input $O_3$, the NO + OH reaction rate is comparable to NO + $O_3$. At 50 ppm input $O_3$, the NO + $NO_3$ reaction rate exceeds the NO + $O_3$ reaction rate, to the point that operation at such high $O_3$ is not advisable. Thus, NO production and loss are not always exactly proportional to $O_3$.

**Revision to Section 3.1 text.** "At lower [$N_2O$], increasing [$O_3$] from 0.5 to 5 ppm increases [NO] because greater NO production from higher [O(1D)] offsets greater NO loss from reaction with OH at 0.5 ppm O3. Increasing [$O_3$] from 5 to 50 ppm decreases [NO] because greater NO loss from reaction with $NO_3$ at 50 ppm O3 offsets greater NO production from higher [O(1D)].

**Revision to Supplement.** Revised Figure S2 is shown below.

[Figure]

9, Figures 2 and 3: according to Hyttinen et al. (2015), high HNO3 concentrations can significantly bias the sensitivities of NO3- CIMS to different highly oxidized compounds. If calibrations for CIMS data were not accordingly performed, CIMS signals may not be considered proportional to concentrations and it is better to not show signal pie charts, which implies the proportionality, and to highlight this caveat in the text.

**Response**. We removed the pie charts from Figures 2 and 3

**Revision to Section 3.2.** Figure 2 shows $NO_3$-CIMS mass spectra […]To examine changes in relative contributions of $C_4H_{4,6,8}O_{4-7}$ , $C_5H_{6,8,10,12}O_{3-8}$, $C_5H_{7,9,11}NO_{6-11}$ , and $C_5H_{10}N_2O_{8-10}$ ions as a function of added $NO_x$, we made two simplifying assumptions: (1) the $NO_3$-CIMS had the same sensitivity to all species that were detected, and (2) $HNO_3$ generated in the reactor did not alter the relative selectivity of the CIMS to different classes of oxidation products, which may not be the case (Hyttinen et al., 2015)."

**Revision to Section 3.3**. "Figure 3 shows $NO_3$-CIMS mass spectra of products generated from the oxidation of α-pinene ($C_{10}H_{16}$). […] As was the case with isoprene oxidation products, we assumed […] the $NO_3$-CIMS had the same sensitivity to all species that were detected, and (3) $HNO_3$ generated in the reactor did not alter the relative selectivity of the CIMS to different classes of oxidation products (Hyttinen et al., 2015)."

**Revision to References**. Added citation for:

Hyttinen, N., Kupiainen-Määttä, O., Rissanen, M. P., Muuronen, M., Ehn, M. and Kurtén, T.: Modeling the Charging of Highly Oxidized Cyclohexene Ozonolysis Products Using Nitrate-Based Chemical Ionization, J. Phys. Chem. A, 119(24), 6339–6345, doi:10.1021/acs.jpca.5b01818, 2015.

10. Page 1, Line 17: it would be better to use the term "highly oxidized molecules (HOM)" instead of "ELVOC", as highly oxidized species may not have extremely low volatility (Kurtén et al., 2016).

**Revision to Introduction**. "Recent atmospheric observations supported by experimental and theoretical studies show that highly oxidized molecules (HOM), together with sulfuric acid, are involved in the initial nucleation steps leading to new particle formation (NPF) (Donahue et al., 2013; Riccobono et al., 2014; Kurtén et al., 2016)."

Revision to References: Added citation for:

Kurtén, T., Tiusanen, K., Roldin, P., Rissanen, M. P., Luy, J.-N., Boy, M., Ehn, M. and Donahue, N. M.: α-pinene Autoxidation Products May Not Have Extremely Low Saturation Vapor Pressures Despite High O:C Ratios, J. Phys. Chem. A, doi:10.1021/acs.jpca.6b02196, 2016.

11. Page 3, Line 16: "condensible" should be "condensable"

**Revision to Section 2**. "These mixing ratios are a factor of 3 to 10 lower than mixing ratios that are typically required to induce homogenous nucleation of condensable oxidation products in related oxidation flow reactor studies (Lambe et al., 2011b)."

12. Page 5, Line 4: "+M" should be added on both sides of the chemical equation.

**Revision to Section 2.3**: "NO+OH + M →HONO + M"

13.        Page 9, Line 4: a space needed between "into" and "C5H7O6-11NO3"

**Revision to Section 3.3.1**. This change will be incorporated in the revised manuscript.

**References**

Z. Peng, D. A. Day, A.M. Ortega, B.B. Palm, W. Hu, H. Stark, R. Li, K. Tsigaridis, W.H. Brune, and J.L. Jimenez. Non-OH chemistry in oxidation flow reactors for the study of atmospheric chemistry systematically examined by modeling. Atmospheric Chemistry and Physics, 16, 4283-4305, doi:10.5194/acp-16-4283-2016, 2016.

Xiong, F., McAvey, K. M., Pratt, K. A., Groff, C. J., Hostetler, M. A., Lipton, M. A., Starn, T. K., Seeley, J. V., Bertman, S. B., Teng, A. P., Crounse, J. D., Nguyen, T. B., Wennberg, P. O., Misztal, P. K., Goldstein, A. H., Guenther, A. B., Koss, A. R., Olson, K. F., De Gouw, J. A., Baumann, K., Edgerton, E. S., Feiner, P. A., Zhang, L., Miller, D. O., Brune, W. H., and Shepson, P. B.: Observation of isoprene hydroxynitrates in the southeastern United States and implications for the fate of NOx, Atmospheric Chemistry and Physics, 15,11, 257–11 272, 2015.